# GyroAtt: A Gyro Attention Framework for Matrix Manifolds

## Abstract

Deep neural networks operating on non-Euclidean geometries, such as Riemannian manifolds, have recently demonstrated impressive performance across various machine-learning applications. Motivated by the success of the attention mechanism, several works have extended it to different geometries. However, existing Riemannian attention methods are mostly designed in an *ad hoc* manner, *i.e.*, tailored to a selected few geometries. Recent studies, on the other hand, show that several matrix manifolds, such as Symmetric Positive Definite (SPD), Symmetric Positive Semi-Definite (SPSD), and Grassmannian manifolds, admit gyro structures, offering a principled way to build Riemannian networks. Inspired by this, we propose a Gyro Attention (GyroAtt) framework over general gyro spaces, applicable to various matrix manifolds. Empirically, we manifest our framework on three gyro structures in the SPD manifold, three in the SPSD manifold, and one in the Grassmannian manifold. Extensive experiments on four electroencephalography (EEG) datasets demonstrate the effectiveness of the proposed framework.

## 1 Introduction

Recently, Deep Neural Networks (DNNs) over Riemannian manifolds, known as Riemannian neural networks, have garnered increasing attention in various applications (Huang & Van Gool, 2017; Gulcehre et al., 2018; Brooks et al., 2019; Shimizu et al., 2021; Kobler et al., 2022; Chen et al., 2023; Wang et al., 2024b; Nguyen et al., 2024; Wang et al., 2024a; Chen et al., 2024e). Commonly encountered manifolds include vector manifolds, such as hyperbolic (Ungar, 2005b) and spherical spaces (Thurston, 1997), and matrix manifolds, such as Symmetric Positive Definite (SPD) (Arsigny et al., 2005), Symmetric Positive Semi-Definite (SPSD) (Bonnabel & Sepulchre, 2010; Bonnabel et al., 2013), and Grassmannian manifolds (Absil et al., 2004). Among these non-Euclidean spaces, hyperbolic manifolds stand out due to the rich algebraic structure of gyrovector spaces (Ungar, 2002; 2005b; 2014), which enables principled and convenient extensions of Euclidean deep learning to hyperbolic manifolds (Gulcehre et al., 2018; Shimizu et al., 2021; Bdeir et al., 2024). In contrast, matrix manifolds provide a compelling balance between structural richness and computational feasibility (Cruceru et al., 2021). Consequently, neural networks on matrix manifolds have emerged as appealing alternatives to their hyperbolic counterparts in various applications (Kim, 2020; Nguyen, 2022b; Nguyen & Yang, 2023; Chen et al., 2024a; Ju et al., 2024). Recently, Kim (2020); Nguyen (2022a;b); Nguyen & Yang (2023) have demonstrated that several matrix manifolds, including SPD, SPSD, and Grassmannian, admit gyrovector space structures, enabling the extension of several fundamental building blocks to matrix manifolds (Nguyen et al., 2024).

Inspired by the great success of the attention mechanism in DNNs (Vaswani et al., 2017; Hu et al., 2018; Dosovitskiy, 2020), researchers have developed attention operations on different geometries. Gulcehre et al. (2018) introduced an attention mechanism for hyperbolic spaces based on the hyperboloid and Klein models, while Pan et al. (2022) extended the attention mechanism to SPD manifolds under Log-Euclidean Metric (LEM). Subsequently, Wang et al. (2024a) further adapted it to Grassmannian manifolds using an extrinsic approach under the projective perspective. However, these designs are tailored for specific manifolds and metrics, limiting their applicability.

As self-attention serves as the prototype of other attention variants, this paper focuses on self-attention. Given that several matrix manifolds admit gyrovector structures, we propose a general framework for attention over gyrovector spaces, called GyroAtt. Unlike previous Riemannian attention approaches, which are tailored to specific geometries (Gulcehre et al., 2018; Pan et al., 2022; Wang et al., 2024a), GyroAtt can be applied across different matrix geometries. Additionally,

GyroAtt naturally generalizes several basic attention blocks to manifolds, including linear transformations, attention computation, and feature aggregation. Specifically, we introduce *gyro homomorphisms*, which extend linear transformations to gyro spaces. The attention mechanism is computed via a score function based on geodesic distances, while aggregation is performed using the weighted Fréchet mean, the manifold counterpart of the Euclidean weighted average. Empirically, we demonstrate the GyroAtt framework on three gyro structures in the SPD manifold, three in the SPSD manifold, and one in the Grassmannian manifold. In summary, our **main contributions** are:

- **Generalizing the attention mechanism to gyrovector spaces.** We propose a principled framework for attention mechanisms over general gyrovector spaces, called GyroAtt. Our method provides a way to directly vary the geometry under the same network structure without constructing manifold-specific operations.

- **Implementation on seven matrix gyrovector spaces.** We implement the GyroAtt framework across three different manifolds: *three gyro structures on the SPD manifold, one on the Grassmannian manifold, and three on the SPSD manifold.*

- **Empirical validation on EEG tasks.** We validate the effectiveness of the proposed GyroAtt framework through experiments on four benchmark EEG datasets. Apart from the superior performance of our GyroAtt, the optimal geometries vary across different tasks, demonstrating the efficacy and flexibility of our GyroAtt framework.

The rest of the paper is organized as follows. Sec. 2 introduces the essential background of gyrovector spaces and Riemannian manifolds. Section 3 examines existing manifold attention mechanisms. We then present our general GyroAtt framework in Sec. 4, detailing its application to various matrix manifolds in Sec. 5. Finally, in Sec. 6, we validate our proposed model on four benchmark EEG datasets. Sec. 7 conclude this paper.

## 2 PRELIMINARY

In this section, we briefly review gyrovector spaces and the concrete gyrovector spaces in the SPD, Grassmannian, and SPSD manifolds. For more in-depth discussions, please refer to Ungar (2005b; 2014); Pennec et al. (2006); Arsigny et al. (2005); Bonnabel et al. (2013); Bendokat et al. (2024).

### 2.1 GYROGROUPS AND GYROVECTOR SPACES

Gyrogroups and gyrovector spaces generalize groups and vector spaces, offering a powerful framework to analyze non-Euclidean geometries. Below, we formally present their definitions.

**Definition 2.1** (**Gyrogroups (Ungar, 2014)**). A gyrogroup is a generalization of groups. Let $G$ be a nonempty set with a binary operation $\oplus$ and an identity element $\mathbf{E} \in G$. Then, a pair $(G, \oplus)$ is a gyrogroup if it satisfies the following axioms:

(G1) There exists an identity element $\mathbf{E} \in G$ such that for all $\mathbf{A} \in G$, $\mathbf{E} \oplus \mathbf{A} = \mathbf{A}$.

(G2) For each $\mathbf{A} \in G$, there exists a left inverse $\ominus\mathbf{A} \in G$ satisfying $\ominus\mathbf{A} \oplus \mathbf{A} = \mathbf{E}$.

(G3) For all $\mathbf{A}, \mathbf{B}, \mathbf{C} \in G$, there exists an automorphism $\mathrm{gyr}[\mathbf{A}, \mathbf{B}](\cdot) : G \to G$, satisfying

$$\mathbf{A} \oplus (\mathbf{B} \oplus \mathbf{C}) = (\mathbf{A} \oplus \mathbf{B}) \oplus \mathrm{gyr}[\mathbf{A}, \mathbf{B}](\mathbf{C}). \tag{1}$$

Here, the map $\mathrm{gyr}[\mathbf{A}, \mathbf{B}](\cdot)$ is called the gyroautomorphism, or the gyration of $G$ generated by $\mathbf{A}, \mathbf{B}$.

(G4) For all $\mathbf{A}, \mathbf{B} \in G$, The map $\mathrm{gyr}[\mathbf{A}, \mathbf{B}]$ generated by each $\mathbf{A}, \mathbf{B}$ satisfies the left loop property: $\mathrm{gyr}[\mathbf{A}, \mathbf{B}] = \mathrm{gyr}[\mathbf{A} \oplus \mathbf{B}, \mathbf{B}]$.

**Definition 2.2** (**Gyrocommutative Gyrogroups Ungar (2014)**). A gyrogroup $(G, \oplus)$ is gyrocommutative if it satisfies the gyrocommutative law: $\mathbf{A} \oplus \mathbf{B} = \mathrm{gyr}[\mathbf{A}, \mathbf{B}](\mathbf{B} \oplus \mathbf{A})$ for all $\mathbf{A}, \mathbf{B} \in G$.

The following definition of gyrovector spaces is derived from Nguyen (2022b, Def. 2.3), which is slightly different from in Ungar (2014, Def. 3.2).

**Definition 2.3** (**Gyrovector Spaces (Nguyen, 2022b)**). A gyrocommutative gyrogroup $(G, \oplus)$ equipped with a scalar multiplication $\otimes : \mathbb{R} \times G \to G$ is a gyrovector space if the following axioms are satisfied:

(V1) $1 \otimes \mathbf{A} = \mathbf{A}, 0 \otimes \mathbf{A} = t \otimes \mathbf{E} = \mathbf{E}$, and $(-1) \otimes \mathbf{A} = \ominus\mathbf{A}$.

(V2) $(s + t) \otimes \mathbf{A} = s \otimes \mathbf{A} \oplus t \otimes \mathbf{A}$.

(V3) $(st) \otimes \mathbf{A} = s \otimes (t \otimes \mathbf{A})$.

(V4) $\mathrm{gyr}[\mathbf{A}, \mathbf{B}](t \otimes \mathbf{C}) = t \otimes \mathrm{gyr}[\mathbf{A}, \mathbf{B}]\mathbf{C}$.

(V5) $\mathrm{gyr}[s \otimes \mathbf{A}, t \otimes \mathbf{A}] = \mathrm{Id}$, where $\mathrm{Id}$ is the identity map.

## 2.2 SPD, GRASSMANNIAN, AND SPSD MANIFOLDS

Table 1: Summary of the gyro additions and geodesic distances over different manifolds.

| Manifold | Metric | Gyro addition | Geodesic distance |
|---|---|---|---|
| SPD | AIM
LEM
LCM | $\mathbf{P} \oplus_{ai} \mathbf{Q} = \mathbf{P}^{\frac{1}{2}} \mathbf{Q} \mathbf{P}^{\frac{1}{2}}$
$\mathbf{P} \oplus_{le} \mathbf{Q} = \mathrm{expm}(\mathrm{logm}(\mathbf{P}) + \mathrm{logm}(\mathbf{Q}))$
$\mathbf{P} \oplus_{lc} \mathbf{Q} = \mathscr{L}^{-1}(\lfloor \mathscr{L}(\mathbf{P}) \rfloor + \lfloor \mathscr{L}(\mathbf{Q}) \rfloor + \mathbb{D}(\mathscr{L}(\mathbf{P}))\mathbb{D}(\mathscr{L}(\mathbf{Q})))$ | $\left\| \mathrm{logm}\left(\mathbf{Q}^{-\frac{1}{2}} \mathbf{P} \mathbf{Q}^{-\frac{1}{2}}\right) \right\|_{\mathbf{F}}$
$\left\| \mathrm{logm}(\mathbf{P}) - \mathrm{logm}(\mathbf{Q}) \right\|_{\mathbf{F}}$
$\left\| \psi_{\mathrm{LC}}(\mathbf{P}) - \psi_{\mathrm{LC}}(\mathbf{Q}) \right\|_{\mathbf{F}}$ |
| Grassmannian | ONB perspective | $\mathbf{U} \widetilde{\oplus}_{gr} \mathbf{V} = \mathrm{expm}([\mathrm{Log}_{\mathbf{I}_{d,q}}^{gr}(\mathbf{U}\mathbf{U}^{\top}), \mathbf{I}_{d,q}])\mathbf{V}$ | $\| \arccos(\Sigma) \|$
$\mathbf{U}^{\top}\mathbf{V} \overset{\mathrm{SVD}}{:=} \mathbf{O}\Sigma\mathbf{R}^{\top}$ |
| SPSD | $(g_{\mathrm{gr}}, \lambda g_{\mathrm{spd}})$ | $(\mathbf{U}_P, \mathbf{S}_P) \oplus_{psd,g} (\mathbf{U}_Q, \mathbf{S}_Q) = (\mathbf{U}_P \widetilde{\oplus}_{gr} \mathbf{U}_Q, \mathbf{S}_P \oplus_g \mathbf{S}_Q)$ | $\mathrm{d}_{\mathrm{gr}}(\mathbf{U}_P, \mathbf{U}_Q) + \lambda \mathrm{d}_{\mathrm{spd}}^g(\mathbf{S}_P, \mathbf{S}_Q)$ |

**SPD manifolds.** Let $\mathcal{S}_d^{++}$ denote the set of $d \times d$ SPD matrices, defined as $\mathcal{S}_d^{++} := \{\mathbf{X} \in \mathbb{R}^{d \times d} \mid \mathbf{X} = \mathbf{X}^{\top}, \mathbf{v}^{\top}\mathbf{X}\mathbf{v} > 0, \forall \mathbf{v} \in \mathbb{R}^d \setminus \{0_d\}\}$. When endowed with a Riemannian metric, $\mathcal{S}_d^{++}$ forms a manifold known as the SPD manifold. Various Riemannian metrics have been introduced on SPD manifolds. In this study, we focus on three prevalent metrics: the Log-Euclidean Metric (LEM) (Arsigny et al., 2005), the Affine-Invariant Metric (AIM) (Pennec et al., 2006), and the Log-Cholesky Metric (LCM) (Lin, 2019). As shown in Nguyen (2022a), these metrics induce corresponding gyrovector spaces—LE, AI, and LC—with the binary operations denoted as $\oplus_{le}$, $\oplus_{ai}$, and $\oplus_{lc}$, and their geodesic distance $\mathrm{d}_{\mathrm{spd}}^{le}(\cdot)$, $\mathrm{d}_{\mathrm{spd}}^{ai}(\cdot)$, and $\mathrm{d}_{\mathrm{spd}}^{lc}(\cdot)$ given by Tab. 1. Here, $\mathbf{P}, \mathbf{Q} \in \mathcal{S}_d^{++}$, $\mathrm{logm}(\cdot)$ and $\mathrm{expm}(\cdot)$ are the matrix logarithm and exponential, respectively. $\mathscr{L}(\mathbf{P})$ represents the Cholesky decomposition of $\mathbf{P}$, yielding a lower triangular matrix with positive diagonal elements such that $\mathbf{P} = \mathscr{L}(\mathbf{P})\mathscr{L}(\mathbf{P})^{\top}$. $\lfloor \mathscr{L}(\mathbf{P}) \rfloor$ denotes the strictly lower triangular part of $\mathscr{L}(\mathbf{P})$, where $\lfloor \mathscr{L}(\mathbf{P}) \rfloor_{(i,j)} = \mathscr{L}(\mathbf{P})_{(i,j)}$ if $i > j$, and zero otherwise. $\mathscr{L}^{-1}(\cdot)$ is the inverse of Cholesky decomposition. $\mathbb{D}(\mathbf{P})$ returns diagonal matrices, where $\mathbb{D}(\mathbf{P})_{(i,i)} = \mathbf{P}_{(i,i)}$.

**Grassmannian manifolds.** The Grassmannian manifold consists of all $q$-dimensional linear subspaces within $\mathbb{R}^d$. Points on the Grassmannian manifold have different matrix representations under various perspectives (Bendokat et al., 2024). In this study, we center on the Orthonormal Basis (ONB) perspective. For clarity, we denote points in the ONB and projector perspective as $\mathbf{Y} \in \widetilde{\mathcal{G}}(q, d)$. In the ONB perspective, a linear subspace is expressed by its orthonormal basis $\mathbf{Y} \in \mathbb{R}^{d \times q}$, where $\mathbf{Y}^{\top}\mathbf{Y} = \mathbf{I}_q$ and $\mathbf{I}_q$ is the $q \times q$ identity matrix. Thus, points on the Grassmannian manifold are equivalence classes of orthonormal bases:

$$[\mathbf{Y}] = \{\widetilde{\mathbf{Y}} \mid \widetilde{\mathbf{Y}} = \mathbf{Y}\mathbf{O}, \mathbf{O} \in \mathrm{O}(q)\}. \tag{2}$$

By abuse of notation, we use $[\mathbf{Y}]$ or $\mathbf{Y}$ interchangeably. As shown by Nguyen & Yang (2023), Grassmannian manifolds in the ONB perspective form nonreductive gyrovector spaces. The binary operation $\widetilde{\oplus}_{gr}$ and geodesic distance $\mathrm{d}_{gr}(\cdot)$ for $\mathbf{U}, \mathbf{V} \in \widetilde{\mathcal{G}}(q, d)$ are defined in Tab. 1. Here, $\mathbf{I}_{d,q} = \begin{bmatrix} \mathbf{I}_q & 0 \\ 0 & 0 \end{bmatrix} \in \mathbb{R}^{d,d}$, $\widetilde{\mathbf{I}}_{d,q} = \begin{bmatrix} \mathbf{I}_q \\ 0 \end{bmatrix} \in \mathbb{R}^{d \times q}$, $[\cdot, \cdot]$ denotes the matrix commutator, and $\mathrm{Log}_{\mathbf{I}_{d,q}}^{gr}$ is the Grassmannian logarithmic map at $\mathbf{I}_{d,q}$ in the projector perspective (details are App. C).

**SPSP manifolds.** The set of $d \times d$ SPSD matrices with rank $q \leq d$ is denoted as $\mathcal{S}_{d,q}^+$. For any $\mathbf{P} \in \mathcal{S}_{d,q}^+$, we decompose it as $\mathbf{P} = \mathbf{U}_P \mathbf{S}_P \mathbf{U}_P^{\top}$, where $\mathbf{U}_P \in \widetilde{\mathcal{G}}(q, d)$ and $\mathbf{S}_P \in \mathcal{S}_d^{++}$ (Bonnabel & Sepulchre, 2010; Bonnabel et al., 2013). Nguyen et al. (2024) introduced a canonical representation of $\mathbf{P}$ in the structure space $\mathcal{G}(q, d) \times \mathcal{S}_q^{++}$. We follow this approach to derive the canonical representation of each point in $\mathcal{S}_{d,q}^+$. Detailed computations are provided in App. E. Based on the above decomposition, we obtain a canonical representation in structure space $(\mathbf{U}_P, \mathbf{S}_P) \in \widetilde{\mathcal{G}}(q, d) \times \mathcal{S}_q^{++}$. When $\mathcal{S}_q^{++}$ is endowed with different Riemannian metrics, it forms distinct nonreductive gyrovector spaces. We use the subscript $g \in \{ai, le, lc\}$ to denote the Riemannian metric on SPD manifolds. Accordingly, the binary operation $\oplus_{psd,g}$ and geodesic distance $\mathrm{d}_{psd,g}$ are defined in Tab. 1, the subscript $g \in \{ai, le, lc\}$ to denote the Riemannian metric on SPD manifolds, $\lambda > 0$.

**Weighted Fréchet mean.** The Weighted Fréchet Mean (WFM) of a set of points $\{\mathbf{P}_{i...N}\}$ on a Riemannian manifold $\mathcal{M}$ is defined as the point $\mathbf{S} \in \mathcal{M}$ that minimizes the weighted sum of squared geodesic distances to all points $\{\mathbf{P}_{i...N}\}$. Given weights $\{w_{1...N}\}$ satisfying the convexity constraint, *i.e.*,$\forall i, w_i > 0$ and $\sum_i w_i = 1$, the WFM is expressed as:

$$\text{WFM}(\{w_i\}, \{\mathbf{P}_i\}) = \underset{\mathbf{S} \in \mathcal{M}}{\arg\min} \sum_{i=1}^{N} w_i \, \mathrm{d}^2\left(\mathbf{P}_i, \mathbf{S}\right), \tag{3}$$

where $\mathrm{d}(\mathbf{P}_i, \mathbf{S})$ is the geodesic distance between the points $\mathbf{S}$ and $\mathbf{P}_i$.

# 3 REVISITING ATTENTION MECHANISMS ON DIFFERENT GEOMETRIES

Table 2: Summary of attention methods on different geometries, where $f_s(\cdot)$ denotes the softmax.

| Method | Geometries | Transformations | $\mathrm{d}(\mathbf{q}_i, \mathbf{k}_i)$ | Attention $\mathcal{A}_{ij}$ | Aggregation $\mathbf{r}_i(\mathbf{R}_i)$ |
|---|---|---|---|---|---|
| Transformer (Vaswani et al., 2017) | Euclidean | $\text{Linear}(\mathbf{x}_i)$ | $\|\mathbf{q}_i - \mathbf{k}_j\|_{\mathbf{F}}$ | $f_s(\langle \mathbf{q}_i, \mathbf{k}_j\rangle / \sqrt{d_k})$ | Arithmetic mean $\sum_j^N \mathcal{A}_{ij}\mathbf{v}_j$ |
| HAN (Gulcehre et al., 2018) | Hyperbolic | $\pi_{\mathbb{R}\to\mathbb{H}}(\text{Linear}(\mathbf{x}_i))$ $\pi_{\mathbb{R}\to\mathbb{K}}(\text{Linear}(\mathbf{x}_i))$ | $\text{arccosh}(-\langle \mathbf{q}_i, \mathbf{k}_i\rangle_M)$ | $f_s(-\beta\,\mathrm{d}(\mathbf{q}_i, \mathbf{k}_j) - c)$ | Einstein midpoint $\sum_j^N \left[\frac{\mathcal{A}_{ij}\gamma(\mathbf{v}_j)}{\sum_l^N \mathcal{A}_{ij}\gamma(\mathbf{v}_l)}\right]\mathbf{v}_j$ |
| MAtt (Pan et al., 2022) | SPD under LEM | $\mathbf{W}\mathbf{X}_i\mathbf{W}^\top$ | $\|\text{logm}(\mathbf{Q}_i) - \text{logm}(\mathbf{K}_i)\|_{\mathbf{F}}$ | $f_s\left((1 + \log(1 + \mathrm{d}(\mathbf{Q}_i, \mathbf{K}_j)))^{-1}\right)$ | LEM-based WFM $\text{expm}\left(\sum_j^N \mathcal{A}_{ij}\text{logm}(\mathbf{V}_j)\right)$ |
| GDLNet (Wang et al., 2024a) | Grassmannian under ONB | $\text{ReOrth}(\mathbf{W}\mathbf{X}_i)$ | $\|\mathbf{Q}_i\mathbf{Q}_i^\top - \mathbf{K}_j\mathbf{K}_j^\top\|_{\mathbf{F}}$ | $f_s\left((1 + \log(1 + \mathrm{d}(\mathbf{Q}_i, \mathbf{K}_j)))^{-1}\right)$ | Extrinsic WFM $\Phi^{-1}\left(\sum_j^N \mathcal{A}_{ij}\Phi(\mathbf{V}_j)\right)$ |
| GyroAtt | Gyro spaces (SPD, SPSD, Grassmann) | Homomorphism Eq. (7) | Geodesic distance | $f_s\left((1 + \log(1 + \mathrm{d}(\mathbf{Q}_i, \mathbf{K}_j)))^{-1}\right)$ | WFM |

The attention mechanism has become a fundamental component in Euclidean deep learning (Vaswani et al., 2017), prompting researchers to extend it to manifolds. A typical attention block comprises three basic units: linear transformation, attention computation, and feature aggregation. Below, we review several Riemannian representatives and summarize the comparison in Tab. 2.

**Euclidean.** Following Vaswani et al. (2017), let $\mathbf{X}, \mathbf{Q}, \mathbf{K}, \mathbf{V}$ and $\mathbf{R}$ represent sets of input features, queries, keys, values, and output features, respectively, and $\mathbf{x}_i, \mathbf{q}_i, \mathbf{k}_i, \mathbf{v}_i$, and $\mathbf{r}_i$ denote $i$-th rows of the corresponding matrices. The feature transformation is performed by a linear map, $\text{Linear}(\cdot)$. Attention is computed as $\text{Softmax}(\langle \mathbf{q}_i, \mathbf{k}_j\rangle / \sqrt{d_k})$, where $\langle \cdot, \cdot \rangle$ denotes the Frobenius inner product and $d_k$ is the dimension of the keys. feature aggregation is defined as $\sum_{j=1}^{N} \mathcal{A}_{ij}\mathbf{v}_j$, where $N$ is the number of values. Generally speaking, the self-attention block requires three basic blocks: 1). a linear transformation to generate $\mathbf{q}_i, \mathbf{k}_i, \mathbf{v}_i$; 2). a correlation- or similarity-based attention for each pair of $\{\mathbf{v}_i, \mathbf{v}_j\}$; 3). the aggregation of the attention-weighted features.

**Hyperbolic.** Gulcehre et al. (2018) introduced the Hyperbolic Attention Network (HAN), a self-attention mechanism for hyperbolic spaces. HAN employs the hyperboloid $\mathbb{H}^d$ and Klein models $\mathbb{K}^d$ of hyperbolic space. Points in the Klein model are obtained by projecting points in Euclidean via $\pi_{\mathbb{R}\to\mathbb{K}}(\cdot)$. The mapping $\pi_{\mathbb{R}\to\mathbb{H}}(\cdot)$ converts Euclidean points to the hyperboloid model using pseudo-polar coordinates. HAN generates attention using $-\beta\,\mathrm{d}(\mathbf{q}_i, \mathbf{k}_j) - c$, where $\beta$ and $c$ are parameters, and employs the Einstein midpoint (Ungar, 2005a) for aggregation.

**SPD manifolds.** Pan et al. (2022) proposed a self-attention mechanism for SPD manifolds under LEM. Considering the input as a set of SPD matrices, we denote $\mathbf{X}_i, \mathbf{Q}_i, \mathbf{K}_i, \mathbf{V}_i, \mathbf{R}_i$ as the input features, queries, keys, values, and output features, respectively. The transformation is applied as $\mathbf{W}\mathbf{X}_i\mathbf{W}^\top$, where $\mathbf{X}_i \in \mathbb{R}^{d_1 \times d_1}$, $\mathbf{W} \in \mathbb{R}^{d_2 \times d_1}$ with $d_1 > d_2$, and $\mathbf{W}$ is semi-orthogonal. Attention is computed as $\text{Softmax}\left((1 + \log(1 + \mathrm{d}(\mathbf{Q}_i, \mathbf{K}_j)))^{-1}\right)$ with $\mathrm{d}(\cdot)$ denotes the LEM-based geodesic distance. Aggregation uses the LEM-based WFM.

**Grassmannian manifolds.** Wang et al. (2024a) proposed a self-attention mechanism for Grassmannian manifolds. By abuse of notation, we use a similar notation to the SPD cases. GDLNet applies the transformation $\text{ReOrth}(\mathbf{W}\mathbf{X}_i)$ (Huang et al., 2018), where $\mathbf{X}_i \in \mathbb{R}^{d_1 \times q}$, $\mathbf{W} \in \mathbb{R}^{d_2 \times d_1}$ with $d_1 > d_2$, $\mathbf{W}$ is semi-orthogonal, $q$ is the dimension of the linear subspaces, and $\text{ReOrth}(\cdot)$ is defined as $\text{ReOrth}(\mathbf{W}\mathbf{X}_i) = \boldsymbol{\Omega}$, with $\mathbf{W}\mathbf{X}_i = \boldsymbol{\Omega}\mathbf{U}$ be a QR decomposition of $\mathbf{W}\mathbf{X}_i$. Attention is computed as $\text{Softmax}\left((1 + \log(1 + \mathrm{d}(\mathbf{Q}_i, \mathbf{K}_j)))^{-1}\right)$ with $\mathrm{d}(\cdot)$ denotes the geodesic distance. The extrinsic WFM (Srivastava & Klassen, 2004) is used for aggregation.

In summary, the above Riemannian attention approaches are confined to particular manifolds or metrics, limiting their application to a broader range of geometries.

## 4 ATTENTION MECHANISMS ON GYROVECTOR SPACES

In this section, we extend the basic attention operations to gyrovector spaces. The Euclidean attention mechanism, as described in Tab. 2, consists of three main operations:

1). **Feature transformation.** This generates $\mathbf{q}_i$, $\mathbf{k}_i$, and $\mathbf{v}_i$ through a linear map $\mathrm{Linear}(\cdot)$ : $\mathbb{R}^n \to \mathbb{R}^m$, which preserves the vector structure as a homomorphism over vector spaces:

$$\mathrm{Linear}(\mathbf{z}_1 + \mathbf{z}_2) = \mathrm{Linear}(\mathbf{z}_1) + \mathrm{Linear}(\mathbf{z}_2); \quad \mathrm{Linear}(t\mathbf{z}) = t\,\mathrm{Linear}(\mathbf{z}), \quad (4)$$

for any $\mathbf{z}, \mathbf{z}_1, \mathbf{z}_2 \in \mathbb{R}^n$ and $t \in \mathbb{R}$.

2). **Attention calculation.** This computes the correlation- or similarity-based attention between $\mathbf{Q}_i$ and $\mathbf{K}_j$ for each pair $\{\mathbf{v}_i, \mathbf{v}_j\}$.

3). **Aggregation.** This aggregates all $\mathbf{v}_i$ based on attention weight matrix $\mathcal{A}$.

We now define the gyro counterparts of the three basic operations mentioned above: 1). Transformation through gyro homomorphisms, which preserve the gyrovector space structure; 2). Distance-based attention; 3). Aggregation via geodesic-based WFM.

**Definition 4.1 (Gyro Homomorphisms).** Let $(\mathcal{M}, \oplus_{\mathcal{M}}, \otimes_{\mathcal{M}}) \to (\mathcal{N}, \oplus_{\mathcal{N}}, \otimes_{\mathcal{N}})$ be two (nonreductive) gyrovector spaces. The map $\mathrm{hom}(\cdot) : (\mathcal{M}, \oplus_{\mathcal{M}}, \otimes_{\mathcal{M}}) \to (\mathcal{N}, \oplus_{\mathcal{N}}, \otimes_{\mathcal{N}})$ is a (nonreductive) gyrovector space homomorphism if it satisfies:

$$\mathrm{hom}(\mathbf{A} \oplus_{\mathcal{M}} \mathbf{B}) = \mathrm{hom}(\mathbf{A}) \oplus_{\mathcal{N}} \mathrm{hom}(\mathbf{B}), \quad \forall \mathbf{A}, \mathbf{B} \in \mathcal{M} \quad (5)$$

$$\mathrm{hom}(t \otimes_{\mathcal{M}} \mathbf{A}) = t \otimes_{\mathcal{N}} \mathrm{hom}(\mathbf{A}), \quad \forall \mathbf{A} \in \mathcal{M}, \forall t \in \mathbb{R}. \quad (6)$$

If we only consider (nonreductive) gyrogroups, $(\mathcal{M}, \oplus_{\mathcal{M}})$ and $(\mathcal{N}, \oplus_{\mathcal{N}})$, a map $\mathrm{hom}(\cdot) : (\mathcal{M}, \oplus_{\mathcal{M}}) \to (\mathcal{N}, \oplus_{\mathcal{N}})$ satisfying Eq. (5) is called a (nonreductive) gyrogroup homomorphism, which has been introduced by Suksumran & Wiboonton (2014). By abuse of notations, we call the above homomorphisms collectively gyro homomorphisms.

Obviously, gyro homomorphism naturally generalizes the linear map in the vector space to the gyrovector space. Thus, we use $\mathrm{hom}(\cdot)$ for the feature transformation. For attention, we calculate the correlation between $\mathbf{Q}_i$ and $\mathbf{K}_j$ using their geodesic distance, then map $\mathrm{d}(\mathbf{Q}_i, \mathbf{K}_j)$ to an attention score, as defined in Eq. (8). For aggregation, we resort to WFM based on the geodesic distance. For a set of input $\{\mathbf{X}_{i...N} \in \mathcal{M}\}$, the key operations of Gyro Attention (GyroAtt) are

$$\mathbf{Q}_i = \mathrm{hom}(\mathbf{X}_i), \mathbf{K}_i = \mathrm{hom}(\mathbf{X}_i), \mathbf{V}_i = \mathrm{hom}(\mathbf{X}_i) \quad \text{(transformation)} \quad (7)$$

$$\mathcal{A} = \mathrm{Softmax}\left(\left(1 + \log(1 + \mathrm{d}(\mathbf{Q}_i, \mathbf{K}_j))\right)^{-1}\right) \quad \text{(Attention)} \quad (8)$$

$$\mathbf{R}_i = \mathrm{WFM}\left(\mathcal{A}_i, \mathbf{V}_{i...N}\right) \quad \text{(Aggregation)} \quad (9)$$

Here, $\mathcal{A}_i$ denote $i$-th rows of $\mathcal{A}$, each output $\mathbf{R}_i$ is the WFM of a set of weights $\mathcal{A}_i$ and $\mathbf{V}_{i...N}$.

To enhance the model's expressivity and capture more complex non-Euclidean correlation, we further apply bias and non-linearity after the aggregation step:

$$\phi(\mathbf{R}_i) = \sigma(\mathbf{B} \oplus \mathbf{R}_i), \quad (10)$$

where $\mathbf{B}$ is a bias, $\sigma$ is a power-based nonlinear activation function.

So far, we have all the ingredients to build attention over general gyrovector spaces, as illustrated in Alg. 1.

## 5 GYRO ATTENTION MECHANISMS ON MATRIX MANIFOLDS

In this section, we showcase our GyroAtt in Alg. 1 across various matrix gyrovector spaces, including three SPD gyro spaces, one Grassmannian gyro space, and three SPSD gyro spaces.

---

**Algorithm 1:** Gyro Attention (GyroAtt) over gyrovector spaces

---

**Input**     : A set of manifold-valued features $\{\mathbf{X}_{1\ldots N} \in \mathcal{M}\}$
**Output**   : A set of manifold-valued features $\{\mathbf{R}'_{1\ldots N}\}$

**for** $i \leftarrow 1$ **to** $N$ **do**
    Queries: $\mathbf{Q}_i = \hom(\mathbf{X}_i)$
    Keys: $\mathbf{K}_i = \hom(\mathbf{X}_i)$
    Values: $\mathbf{V}_i = \hom(\mathbf{X}_i)$
**end**
**for** $i \leftarrow 1$ **to** $N$ **do**
    **for** $j \leftarrow 1$ **to** $N$ **do**
        Similarity calculation: $\mathcal{S}_{ij} = (1 + \log(1 + \mathrm{d}(\mathbf{Q}_i, \mathbf{K}_j)))^{-1}$
    **end**
    Attention calculation: $\mathcal{A}_{ij} = \mathrm{Softmax}(\mathcal{S}_{ij})$
    Aggregation: $\mathbf{R}_i = \mathrm{WFM}(\{\mathcal{A}_{ij}\}_{j=1}^N, \{\mathbf{V}_j\}_{j=1}^N)$
    Bias and nonlinearity: $\mathbf{R}'_i = \sigma(\mathbf{R}_i \oplus \mathbf{B})$
**end**

---

## 5.1 GYRO ATTENTION MECHANISMS ON SPD GYROVECTOR SPACES

As shown by Tab. 1, there are three SPD gyrovector spaces, induced by AIM, LEM, and LCM, respectively. The geodesic distance (for attention calculation) and gyro addition (for biasing) have already been well studied over these three metrics (Arsigny et al., 2005; Pennec et al., 2006; Lin, 2019). The operations have been summarized in Tab. 1. We only need to discuss the gyro homomorphisms, WFM, and activation over these three geometries. We first identify the concrete expressions of gyro homomorphism over different SPD gyro spaces. Due to page limitations, the proofs are provided in the App. G and can be accessed by clicking [↓].

**Theorem 5.1** (**AIM Homomorphisms**). [↓] *Let* $\mathbf{P} \in (\mathcal{S}_d^{++}, \oplus_{ai}, \otimes_{ai})$*, and* $\mathbf{O} \in \mathrm{O}(d)$ *be an orthogonal matrix. The transformation map* $\hom_{ai}(\cdot) : (\mathcal{S}_d^{++}, \oplus_{ai}, \otimes_{ai}) \to (\mathcal{S}_d^{++}, \oplus_{ai}, \otimes_{ai})$ *defined by*

$$\hom_{ai}(\mathbf{P}) = \mathbf{O}\mathbf{P}\mathbf{O}^\top, \tag{11}$$

*is a gyro homomorphism.*

**Theorem 5.2** (**LEM Homomorphisms**). [↓] *Let* $\mathbf{P} \in (\mathcal{S}_d^{++}, \oplus_{le}, \otimes_{le})$*, and let* $\mathbf{M} \in \mathbb{R}^{n \times n}$*. The transformation map* $\hom_{le}(\cdot) : (\mathcal{S}_d^{++}, \oplus_{le}, \otimes_{le}) \to (\mathcal{S}_d^{++}, \oplus_{le}, \otimes_{le})$ *defined by*

$$\hom_{le}(\mathbf{P}) = \mathrm{expm}(\mathbf{M}\,\mathrm{logm}(\mathbf{P})\mathbf{M}^\top), \tag{12}$$

*is a gyro homomorphism.*

**Corollary 5.3** (**LEM Homomorphisms**). [↓] *For* $\mathbf{P} \in (\mathcal{S}_d^{++}, \oplus_{le}, \otimes_{le})$*, if* $\mathbf{O} \in \mathrm{O}(d)$ *is an orthogonal matrix, the gyro homomorphism Eq.* (12) *is simplified as*

$$\hom_{le}(\mathbf{P}) = \mathbf{O}\mathbf{P}\mathbf{O}^\top. \tag{13}$$

**Theorem 5.4** (**LCM Homomorphisms**). [↓] *Let* $\mathbf{P} \in (\mathcal{S}_d^{++}, \oplus_{lc}, \otimes_{lc})$*, and let* $\mathbf{M} \in \mathbb{R}^{n \times n}$*. The transformation map* $\hom_{lc}(\cdot) : (\mathcal{S}_d^{++}, \oplus_{lc}, \otimes_{lc}) \to (\mathcal{S}_d^{++}, \oplus_{lc}, \otimes_{lc})$ *defined by*

$$\hom_{lc}(\mathbf{P}) = \mathscr{L}^{-1}\big(\lfloor L(\mathbf{P}) \rfloor + \mathrm{expm}(\mathbb{D}(L(\mathbf{P})))\big), \tag{14}$$

*where*

$$L(\mathbf{P}) = \mathbf{M}\left(\lfloor \mathscr{L}(\mathbf{P}) \rfloor + \lfloor \mathscr{L}(\mathbf{P}) \rfloor^\top + \mathbb{D}(\mathscr{L}(\mathbf{P}))\right)\mathbf{M}^\top, \tag{15}$$

*is a gyro homomorphism.*

**Transformation.** Orthogonal constraints can improve network generalization by serving as implicit regularization (Lezcano-Casado & Martınez-Rubio, 2019). Therefore, we further impose orthogonality on $\mathbf{M}$ in both $\hom_{lc}(\cdot)$ and $\hom_{le}(\cdot)$. Consequently, the involved transformation layers under three metrics are Eq. (11) for AIM, Eq. (13) for LEM, and Eq. (14) for LCM.

**WFMs.** The WFMs under LEM and LCM have closed-form expressions, while the ones under AIM can be computed using the Karcher flow algorithm (Karcher, 1977), an iterative method. (Karcher, 1977). Detailed algorithms for the WFMs under AIM are provided in App. D.1.

Table 3: Key operators in calculating GyroAtt on gyrovector spaces.

| Manifold | SPD | | | Grassmannian | SPSD |
|---|---|---|---|---|---|
| Metric | AIM | LEM | LCM | ONB perspective | $(g_{gr}, \lambda g_{spd})$ |
| Homomorphism | $\mathbf{OPO}^\top$ | $\mathbf{OPO}^\top$ | Eq. (14) | $\mathbf{OU}$ | $(\mathrm{hom}_{gr}(\mathbf{U}_P), \mathrm{hom}_g(\mathbf{S}_P))$ |
| WFM | Karcher flow Alg. A1 | Closed-form Eq. (A17) | Closed-form Eq. (A18) | Karcher flow Alg. A2 | $(\mathrm{WFM}_{spd}, \mathrm{WFM}_{gr})$ |
| Bias and Non-linearity | $(\mathbf{B}_{spd} \oplus_g \mathbf{R}_i)^p$ | | | $\mathbf{B}_{gr} \widetilde{\oplus}_{gr} \mathbf{R}_i$ | $(\mathbf{U}_{R_i}, (\mathbf{S}_{R_i})^p)$ |

**Activation.** As demonstrated by Chen et al. (2024d, Fig. 1) and Chen et al. (2024b, Sec. 5.1), the matrix power can deform the latent SPD geometries. Thus, we use matrix power as the activation function to activate the underlying Riemannian geometry.

## 5.2 GYRO ATTENTION ON GRASSMANNIAN MANIFOLDS

We implement the GyroAtt framework on the ONB Grassmannian nonreductive gyrovector spaces. The geodesic distance and gyro addition are given by Tab. 1. Similar to the SPD gyro spaces, we use gyro homomorphism for transformation and WFM for aggregation. As shown by Nguyen & Yang (2023, Sec. 2.3.2), the Grassmannian gyro addition can be viewed as non-linear activation. Therefore, we do not use additional activation before the Grassmannian gyro biasing. In the following, we discuss gyro homomorphism and WFM over the Grassmannian.

**Theorem 5.5 (Grassmannian Homomorphisms).** [↓] *Let* $\mathbf{U} \in (\widetilde{\mathcal{G}}(q, d), \widetilde{\oplus}_{gr}, \widetilde{\otimes}_{gr})$, *and let* $\mathbf{O} = \begin{bmatrix} \mathbf{O}_q & 0 \\ 0 & \mathbf{O}_{d-q} \end{bmatrix} \in \mathbb{R}^{d,d}$, *where* $\mathbf{O}_q \in \mathbb{R}^{q \times q}$ *and* $\mathbf{O}_{d-q} \in \mathbb{R}^{(d-q) \times (d-q)}$ *are orthogonal matrices. The transformation map* $\mathrm{hom}_{gr}(\cdot) : (\widetilde{\mathcal{G}}(q, d), \widetilde{\oplus}_{gr}, \widetilde{\otimes}_{gr}) \to (\widetilde{\mathcal{G}}(q, d), \widetilde{\oplus}_{gr}, \widetilde{\otimes}_{gr})$ *defined by*

$$\mathrm{hom}_{gr}(\mathbf{U}) = \mathbf{OU}, \tag{16}$$

*is a gyro homomorphism.*

We use Eq. (16) for the Grassmannian feature transformation. For weighted aggregation, since the WFM on the Grassmannian manifold lacks a closed-form solution, we utilize the Karcher flow algorithm (Absil et al., 2004; Karcher, 1977). More details are exposed in App. D.2.

## 5.3 GYRO ATTENTION MECHANISMS ON SPSD MANIFOLDS

As outlined in Sec. 2, any $\mathbf{P} \in \mathcal{S}_{d,q}^+$ can be represented in the structured space as $(\mathbf{U}_P, \mathbf{S}_P) \in \widetilde{\mathcal{G}}(q, d) \times \mathcal{S}_q^{++}$ using the canonical representation. As shown in Tab. 1, the distance and gyro addition in the structured space are defined by the product space. To implement the GyroAtt framework in the SPSD gyrovector space, we only need to show gyro homomorphism, WFM, and activation.

**Theorem 5.6 (SPSD Homomorphisms).** [↓] *Let* $(\mathbf{U}_P, \mathbf{S}_P) \in (\widetilde{\mathcal{G}}(q, d) \times \mathcal{S}_q^{++}, \oplus_{psd,g}, \otimes_{psd,g})$, *with* $g \in \{ai, le, lc\}$. *The transformation map* $\mathrm{hom}_{psd,g}(\cdot) : (\widetilde{\mathcal{G}}(q, d) \times \mathcal{S}_q^{++}, \oplus_{psd,g}, \otimes_{psd,g}) \to (\widetilde{\mathcal{G}}(q, d) \times \mathcal{S}_q^{++}, \oplus_{psd,g}, \otimes_{psd,g})$ *defined by*

$$\mathrm{hom}_{psd,g}(\mathbf{U}_P, \mathbf{S}_P) = (\mathrm{hom}_{gr}(\mathbf{U}_P), \mathrm{hom}_g(\mathbf{S}_P)), \tag{17}$$

*is a gyro homomorphism.*

For the aggregation, we use the WFM by the product of the structured space, detailed in App. D.3. Bias and non-linearity are also defined by product space:

$$\phi_{psd}(\mathbf{U}_{R_i}, \mathbf{S}_{R_i}) = (\mathbf{U}_{R_i}, (\mathbf{S}_{R_i})^p). \tag{18}$$

## 5.4 SUMMARY OF GYROATT IN MATRIX MANIFOLDS

In summary, our GyroAtt framework comprises several basic operations. We begin by applying the mapping $\mathrm{hom}(\cdot)$ to obtain the $\mathbf{Q}_i$, $\mathbf{K}_i$, and $\mathbf{V}_i$. Attention scores are then computed using geodesic distances between these queries and keys. To aggregate the values $\mathbf{V}_i$, we employ the WFM. Finally, we enhance the model's expressive capacity by introducing bias and applying a non-linear activation function. Tab. 3 summarizes all the key ingredients for computing GyroAtt on SPD, Grassmannian, and SPSD manifolds.

Table 4: Average test set results and standard deviation on the MAMEM-SSVEP-II and BCI-ERN datasets. Other Riemannian attention methods are highlighted with a light yellow background. The best three results are highlighted with **red**, **blue**, **cyan**.

| Methods | MAMEM-SSVEP-II | BCI-ERN |
|---|---|---|
| EEGNet (Lawhern et al., 2018) | $53.7 \pm 7.2$ | $74.3 \pm 2.5$ |
| ShallowCNet (Schirrmeister et al., 2017) | $56.9 \pm 6.7$ | $71.9 \pm 2.6$ |
| SCCNet (Wei et al., 2019) | $62.1 \pm 7.7$ | $70.9 \pm 2.3$ |
| EEG-TCNet (Ingolfsson et al., 2020) | $55.5 \pm 7.7$ | $77.1 \pm 2.5$ |
| FBCNet (Mane et al., 2021) | $53.1 \pm 5.7$ | $60.5 \pm 3.1$ |
| TCNet-Fusion (Musallam et al., 2021) | $45.0 \pm 6.6$ | $70.5 \pm 2.9$ |
| MBEEGSE (Altuwaijri et al., 2022) | $56.5 \pm 7.3$ | $75.5 \pm 2.3$ |
| SPDNetBN (Brooks et al., 2019) | $62.8 \pm 3.1$ | $72.3 \pm 3.5$ |
| MAtt (Pan et al., 2022) | $65.2 \pm 3.1$ | $75.7 \pm 2.2$ |
| GDLNet (Wang et al., 2024a) | $65.5 \pm 2.9$ | $78.2 \pm 2.5$ |
| GyroAtt-SPD | $66.3 \pm 2.2$ | $76.1 \pm 4.2$ |
| GyroAtt-Gr | $67.1 \pm 1.6$ | $78.4 \pm 1.4$ |
| GyroAtt-SPSD | $68.7 \pm 1.5$ | $79.1 \pm 1.7$ |

Table 5: Average test set results and standard deviation on the BNCI2014001 and BNCI2015001 datasets. Other Riemannian attention methods are highlighted with a light yellow background. The best three results are highlighted with **red**, **blue**, **cyan**.

| Methods | BNCI2014001 | | BNCI2015001 | |
|---|---|---|---|---|
| | Inter-session | Inter-subject | Inter-session | Inter-subject |
| FBCSP+SVM (Ang et al., 2008) | $60.6 \pm 4.9$ | $32.3 \pm 7.3$ | $81.5 \pm 4.4$ | $58.6 \pm 13.4$ |
| TSM+SVM (Barachant et al., 2011) | $61.8 \pm 4.1$ | $34.7 \pm 8.6$ | $75.7 \pm 5.1$ | $56.0 \pm 6.0$ |
| FB+TSM+LR (Kobler et al., 2021) | $69.8 \pm 4.8$ | $36.5 \pm 8.2$ | $80.9 \pm 6.0$ | $60.6 \pm 10.9$ |
| EEGNet (Lawhern et al., 2018) | $41.8 \pm 5.8$ | $43.3 \pm 17.0$ | $72.4 \pm 8.4$ | $59.2 \pm 9.5$ |
| ShConvNet (Schirrmeister et al., 2017) | $51.3 \pm 2.3$ | $42.2 \pm 16.2$ | $74.1 \pm 4.2$ | $58.7 \pm 5.8$ |
| FBCSP+DSS+LDA (Hehenberger et al., 2021) | $71.3 \pm 1.8$ | $48.3 \pm 14.3$ | $84.6 \pm 4.8$ | $67.7 \pm 14.3$ |
| URPA+MDM (Rodrigues et al., 2018) | $59.5 \pm 2.7$ | $46.8 \pm 14.6$ | $79.2 \pm 4.6$ | $70.3 \pm 16.1$ |
| SPDOT+TSM+SVM (Yair et al., 2019) | $66.8 \pm 3.8$ | $38.6 \pm 8.6$ | $77.5 \pm 2.9$ | $63.3 \pm 8.1$ |
| TSMNet (Kobler et al., 2022) | $69.0 \pm 3.6$ | $51.6 \pm 16.5$ | $85.8 \pm 4.3$ | $77.0 \pm 13.7$ |
| Graph-CSPNet (Ju & Guan, 2023) | $71.9 \pm 13.3$ | $45.2 \pm 9.3$ | $79.8 \pm 14.6$ | $64.2 \pm 13.4$ |
| MAtt (Pan et al., 2022) | $66.5 \pm 8.9$ | $45.3 \pm 11.3$ | $80.8 \pm 14.8$ | $63.1 \pm 10.1$ |
| GDLNet (Wang et al., 2024a) | $58.1 \pm 8.9$ | $46.3 \pm 5.1$ | $76.9 \pm 13.6$ | $63.3 \pm 14.2$ |
| GyroAtt-SPD | $75.4 \pm 7.4$ | $53.1 \pm 14.1$ | $86.2 \pm 4.5$ | $77.9 \pm 13.0$ |
| GyroAtt-Gr | $72.5 \pm 7.3$ | $52.1 \pm 14.2$ | $85.0 \pm 7.7$ | $75.3 \pm 13.7$ |
| GyroAtt-SPSD | $72.9 \pm 6.2$ | $52.4 \pm 15.6$ | $85.3 \pm 5.3$ | $76.0 \pm 14.1$ |

## 6 EXPERIMENTS

In this paper, we evaluate the performance of the proposed Gyro Attention Network in EEG signal classification. Building on prior studies (Pan et al., 2022; Kobler et al., 2022), we evaluate four datasets: BNCI2014001 (Faller et al., 2012), BNCI2015001 (Tangermann et al., 2012), MAMEM-SSVEP-II (Spiros, 2016), and BCI-ERN (Margaux et al., 2012). For the BNCI2014001 and BNCI2015001 datasets, we conduct both inter-session and inter-subject evaluations. For the inter-session evaluation, models are trained exclusively on data from the corresponding subject. The balanced accuracy calculated by the average recall across classes is taken as our performance metric (Kobler et al., 2022). For the MAMEM-SSVEP-II and BCI-ERN datasets, accuracy is used as the evaluation metric for MAMEM-SSVEP-II, while for BCI-ERN, the Area Under the Curve addresses class imbalance. In the experiments, the first four sessions of each subject in each dataset are designated for training, with one session reserved for validation. The network is subsequently tested on the fifth session. App. B.2 introduces all the used datasets and preprocessing steps.

**Implementation details.** The GyroAtt network architecture, depicted in Fig. 1, comprises three main components: a feature extraction module, a Gyro Attention module, and a classification mod-

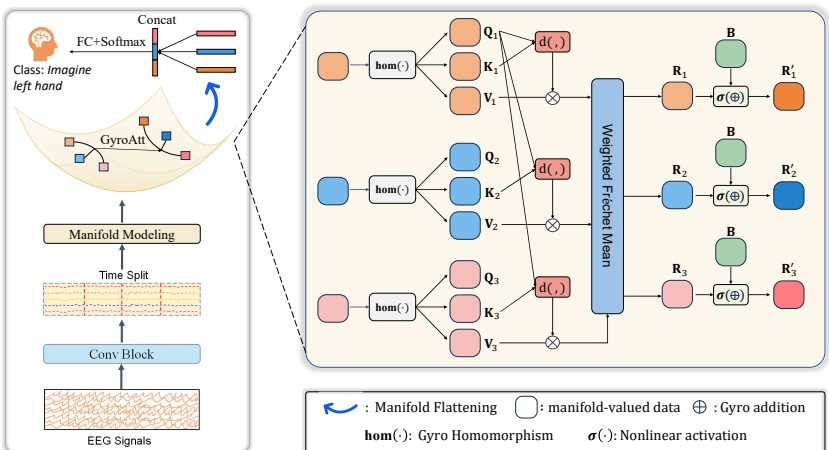

Figure 1: The GyroAtt network architecture comprises three components: a feature extraction module that converts EEG signals into manifold-valued data, a Gyro Attention module that explicitly captures long-range dependencies among features, and a classification module that flattens manifold data before classification using a fully connected layer and a softmax function.

ule. In the feature extraction module, we first apply two convolutional blocks to the EEG signals to extract low-redundancy features. We then perform pyramid-like segmentation along the time dimension on the outputs, partitioning the data into $s$ non-overlapping subparts at each level $s$. For each subpart, a covariance matrix is computed. For GyroAtt-SPD, these covariance matrices $\mathbf{X}_i$ serve directly as inputs to the subsequent layers. In GyroAtt-SPSD and GyroAtt-Gr, each covariance matrix is transformed into its canonical form $(\mathbf{U}_X^i, \mathbf{S}_X^i)$ using Alg. A3, mapping them into the structure space $\widetilde{\mathcal{G}}(q, d) \times \mathcal{S}_q^{++}$. Here, $\mathbf{U}_X^i$ is used as the input for GyroAtt-Gr, while both $\mathbf{U}_X^i$ and $\mathbf{S}_X^i$ are used for GyroAtt-SPSD. We employ the corresponding GyroAtt block, as shown in Alg. 1, to capture long-range dependencies between different feature regions on the manifolds. In the classification module, we first perform manifold flattening by projecting the manifold-valued data into a flat space and vectorizing it. for the GyroAtt-SPD, we apply matrix power normalization to the output matrix $\mathbf{P}$ from the GyroAtt block, defined as $\psi_\theta(\mathbf{P}) = \frac{1}{\theta}\mathbf{P}^\theta$ with $\theta > 0$ and $\mathbf{P} \in \mathcal{S}_d^{++}$, following the approach in Wang et al. (2020); Chen et al. (2024c). The scaling factor $\frac{1}{\theta}$ ensures gradient stability during optimization. For GyroAtt-Gr, we project each element $\mathbf{Y}_i \in \mathcal{G}(q, d)$ into Euclidean space using the operator $\Phi(\mathbf{Y}_i) = \mathbf{Y}_i\mathbf{Y}_i^\top$. In the GyroAtt-SPSD model, both $\mathbf{U}_X^i$ and $\mathbf{S}_X^i$ are processed accordingly within the classification module. Across all three models, the resulting matrices are vectorized, concatenated, and passed through a fully connected layer followed by a $\mathrm{Softmax}$ function for classification. For manifold parameter optimization and detailed implementations for different datasets, please refer to App. F and App. B.3.

**Parameter settings.** We report results using the best settings for each manifold; additional results are provided in Tab. 6. We use the notation {Metric, $p$, $\theta$} to specify parameters—e.g., {AIM, 0.5, 0.5} means the metric is AIM with $p$ and $\theta$ set to 0.5. For GyroAtt-SPD, the settings are: {AIM, 0.75, 0.75} on MAMEM-SSVEP-II, {LEM, 0.75, 0.75} on BCI-ERN, and {AIM, 0.5, 0.5} on both BNCI2014001 and BNCI2015001. For GyroAtt-SPSD, the settings are {LCM, 0.5, 0.5} on MAMEM-SSVEP-II, {LEM, 0.5, 0.25} on BCI-ERN, {AIM, 0.5, 0.5} on BNCI2014001, and {LEM, 0.25, 0.25} on BNCI2015001.

**Main results.** We evaluated the performance of our proposed GyroAtt framework on four EEG classification datasets, with the 10-fold cross-validation results summarized in Tab. 4 and Tab. 5. Our models—GyroAtt-SPD, GyroAtt-Gr, and GyroAtt-SPSD—were compared against other leading methods. The manifold yielding the most effective GyroAtt layer varies across datasets. Specifically, GyroAtt-SPSD provides optimal performance on the MAMEM-SSVEP-II and BCI-ERN datasets, surpassing GDLNet by 3.2% and 0.9%, respectively. GyroAtt-SPD achieves the best results on the BNCI2014001 and BNCI2015001 datasets, outperforming TSMNet by 6.4%, 1.5%, 0.4%, and 0.9%. This finding highlights the versatility of our framework. Although GyroAtt-Gr performs worse than GyroAtt-SPSD on these datasets, it still surpasses GDLNet across all four datasets. These

Table 6: Ablations of GyroAtt on Riemannian metrics and matrix power activation $p$. The best result under each geometry is highlighted in **bold**.

| Geometry | $p$ | BNCI2014001 | | BNCI2015001 | | MAMEM-SSVEP-II |
| | | Inter-session | Inter-subject | Inter-session | Inter-subject | |
|---|---|---|---|---|---|---|
| SPD-AIM | w/o | $74.8 \pm 6.7$ | $51.2 \pm 15.7$ | $85.5 \pm 5.0$ | $75.4 \pm 12.9$ | $64.3 \pm 2.4$ |
| | 0.25 | $75.2 \pm 6.9$ | $51.4 \pm 14.3$ | $85.8 \pm 6.8$ | $77.1 \pm 12.8$ | $61.9 \pm 2.5$ |
| | 0.50 | $\mathbf{75.4 \pm 7.1}$ | $\mathbf{53.1 \pm 14.8}$ | $\mathbf{86.2 \pm 4.5}$ | $\mathbf{77.9 \pm 13.0}$ | $66.1 \pm 2.6$ |
| | 0.75 | $75.0 \pm 8.1$ | $51.0 \pm 13.8$ | $85.9 \pm 6.6$ | $77.4 \pm 12.6$ | $\mathbf{66.3 \pm 2.2}$ |
| SPD-LEM | w/o | $74.9 \pm 7.3$ | $51.7 \pm 15.8$ | $85.2 \pm 5.2$ | $75.3 \pm 12.3$ | $65.6 \pm 2.3$ |
| | 0.25 | $74.7 \pm 6.7$ | $\mathbf{52.3 \pm 14.1}$ | $85.6 \pm 6.7$ | $75.6 \pm 13.0$ | $63.7 \pm 2.5$ |
| | 0.50 | $\mathbf{75.3 \pm 6.5}$ | $51.4 \pm 14.1$ | $\mathbf{85.7 \pm 5.5}$ | $\mathbf{76.6 \pm 13.7}$ | $65.3 \pm 2.7$ |
| | 0.75 | $75.1 \pm 7.3$ | $52.3 \pm 15.0$ | $85.4 \pm 7.0$ | $75.5 \pm 12.8$ | $\mathbf{66.2 \pm 2.5}$ |
| SPD-LCM | w/o | $73.2 \pm 6.7$ | $51.9 \pm 14.8$ | $85.3 \pm 7.2$ | $76.2 \pm 13.3$ | $64.0 \pm 2.8$ |
| | 0.25 | $73.4 \pm 7.5$ | $52.4 \pm 13.4$ | $85.6 \pm 7.5$ | $75.4 \pm 14.0$ | $64.3 \pm 2.5$ |
| | 0.50 | $74.0 \pm 8.2$ | $\mathbf{52.7 \pm 13.6}$ | $85.9 \pm 6.7$ | $\mathbf{77.4 \pm 13.2}$ | $64.1 \pm 3.2$ |
| | 0.75 | $\mathbf{74.2 \pm 7.8}$ | $51.7 \pm 14.6$ | $\mathbf{86.0 \pm 6.8}$ | $76.3 \pm 13.2$ | $\mathbf{65.1 \pm 2.5}$ |
| SPSD-AIM | w/o | $72.2 \pm 7.2$ | $49.2 \pm 13.7$ | $84.1 \pm 7.2$ | $73.6 \pm 14.3$ | $65.8 \pm 2.6$ |
| | 0.25 | $72.4 \pm 6.8$ | $50.7 \pm 14.8$ | $84.0 \pm 6.8$ | $\mathbf{75.5 \pm 13.8}$ | $66.4 \pm 3.0$ |
| | 0.50 | $\mathbf{72.9 \pm 7.1}$ | $\mathbf{52.4 \pm 15.6}$ | $\mathbf{84.7 \pm 6.6}$ | $74.2 \pm 14.2$ | $\mathbf{66.5 \pm 2.9}$ |
| | 0.75 | $72.5 \pm 6.7$ | $51.0 \pm 15.3$ | $84.0 \pm 4.9$ | $74.5 \pm 13.6$ | $65.7 \pm 2.7$ |
| SPSD-LEM | w/o | $72.1 \pm 6.7$ | $49.8 \pm 12.9$ | $83.9 \pm 5.1$ | $74.2 \pm 14.4$ | $66.2 \pm 1.9$ |
| | 0.25 | $\mathbf{72.8 \pm 6.9}$ | $49.9 \pm 14.0$ | $\mathbf{85.3 \pm 5.3}$ | $\mathbf{76.0 \pm 14.1}$ | $\mathbf{66.5 \pm 2.3}$ |
| | 0.50 | $72.5 \pm 6.6$ | $50.5 \pm 14.2$ | $84.5 \pm 5.8$ | $75.4 \pm 14.5$ | $66.4 \pm 2.5$ |
| | 0.75 | $72.7 \pm 7.6$ | $\mathbf{50.5 \pm 13.2}$ | $85.2 \pm 4.8$ | $75.0 \pm 14.2$ | $66.2 \pm 1.7$ |
| SPSD-LCM | w/o | $72.3 \pm 7.3$ | $49.5 \pm 12.0$ | $84.8 \pm 6.1$ | $75.4 \pm 13.2$ | $66.5 \pm 2.4$ |
| | 0.25 | $72.2 \pm 7.5$ | $50.6 \pm 13.9$ | $\mathbf{85.1 \pm 4.8}$ | $\mathbf{74.9 \pm 12.6}$ | $67.7 \pm 2.3$ |
| | 0.50 | $\mathbf{72.9 \pm 6.7}$ | $48.4 \pm 13.3$ | $84.9 \pm 6.1$ | $74.5 \pm 13.6$ | $66.2 \pm 3.6$ |
| | 0.75 | $72.8 \pm 6.3$ | $\mathbf{51.7 \pm 13.1}$ | $85.1 \pm 5.8$ | $73.9 \pm 15.4$ | $\mathbf{68.7 \pm 1.5}$ |

observations highlight the generality and effectiveness of our GyroAtt approach. Furthermore, the superior performance of GyroAtt can be attributed to its attention mechanism, which effectively captures long-range dependencies and spatiotemporal fluctuations inherent in EEG data.

**Ablations on the Riemannian metrics and matrix power-based nonlinear activation $\sigma(\cdot)$ in GyroAtt.** Tab. 6 illustrates the impact of the different metrics and power parameter $p$ (as defined in Tab. 3) on the performance of GyroAtt based on two Riemannian matrix manifolds. The candidate values of metrics are AIM, LEM, and LCM, with $p$ values set to $\{0.25, 0.50, 0.75\}$. As shown in this table, for SPD-based architectures, GyroAtt under the SPD-AIM geometry with $p = 0.5$ achieves the highest accuracy on both the BNCI2014001 and BNCI2015001 datasets, while the SPD-LCM geometry with $p = 0.75$ records the second-highest inter-session accuracy (86.0%) on the BNCI2015001 dataset. For SPSD-based settings, GyroAtt under the SPSD-LCM geometry with $p = 0.75$ reaches the highest accuracy (68.7%) on the MAMEM-SSVEP-II dataset. Furthermore, it is evident that GyroAtt is generally robust to variations in $p$ across all experimental scenarios. These findings emphasize the importance of selecting the metric space of the underlying feature manifold and demonstrate that the proposed matrix power activation enhances model performance by introducing nonlinearity into the metric space.

## 7 CONCLUSION

In this paper, we propose the GyroAtt framework, which extends the Euclidean attention mechanism to general gyrovector spaces in a principled manner. Specifically, we adopt gyro homomorphisms, geodesic-based attention, and WFM as counterparts to the transformation, attention, and aggregation operations in Euclidean attention. Notably, we identify the concrete non-trivial expressions of gyro homomorphisms on different matrix gyro spaces. The principled construction of GyroAtt enables a direct assessment of the impact of geometry on a given task while keeping the neural network architecture constant. Extensive experiments on four EEG datasets demonstrate the efficacy and flexibility of our approach. For future avenues, we will implement our GyroAtt framework on other concrete gyro spaces.

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

# A  NOTATIONS AND ABBREVIATIONS

For better clarity, we summarize all the notations and the abbreviations used in this paper in Tab. A1 and Tab. A2, respectively.

Table A1: Summary of notations.

| Notations | Explanation |
|---|---|
| $(G, \oplus)$ | A gyrogroup $G$ with a binary operation $\oplus$ |
| $\mathcal{S}_d^{++}$ | Space of $d \times d$ SPD matrices |
| $\mathcal{S}^d$ | Space of $d \times d$ symmetric matrices |
| $\mathcal{S}_{d,q}^+$ | Space of $d \times d$ SPSD matrices with rank $q \leq d$ |
| $\mathcal{G}(q, d)$ | Grassmannian in the projector perspective |
| $\widetilde{\mathcal{G}}(q, d)$ | Grassmannian in the ONB perspective |
| $\oplus_{ai}, \ominus_{ai}, \otimes_{ai}$ | Binary, inverse, and scalar multiplication operations in $\mathcal{S}_d^{++}$ under AIM |
| $\oplus_{le}, \ominus_{le}, \otimes_{le}$ | Binary, inverse, and scalar multiplication operations in $\mathcal{S}_d^{++}$ under LEM |
| $\oplus_{lc}, \ominus_{lc}, \otimes_{lc}$ | Binary, inverse, and scalar multiplication operations in $\mathcal{S}_d^{++}$ under LCM |
| $\widetilde{\oplus}_{gr}, \widetilde{\ominus}_{gr}, \widetilde{\otimes}_{gr}$ | Binary, inverse, and scalar multiplication operations in $\widetilde{\mathcal{G}}(q, d)$ |
| $\oplus_{gr}, \ominus_{gr}, \otimes_{gr}$ | Binary, inverse, and scalar multiplication operations in $\mathcal{G}(q, d)$ |
| $\oplus_{psd,g}, \ominus_{psd,g}, \otimes_{psd,g}$ | Binary, inverse, and scalar multiplication operations in $\widetilde{\mathcal{G}}(q, d) \times \mathcal{S}_d^{++}$ under metrics $g$ |
| $\langle \mathbf{P}, \mathbf{Q} \rangle^g$ | Inner product in $\mathcal{S}_d^{++}$ under metrics $g$ |
| $\langle \mathbf{U}, \mathbf{V} \rangle^{gr}$ | Inner product in $\widetilde{\mathcal{G}}(q, d)$ |
| $\langle (\mathbf{U}_P, \mathbf{S}_P), (\mathbf{U}_Q, \mathbf{S}_Q) \rangle^{psd,g}$ | Inner product in $\widetilde{\mathcal{G}}(q, d) \times \mathcal{S}_d^{++}$ under metrics $g$ |
| $\|\ominus_g \mathbf{P} \oplus_g \mathbf{Q}\|_g^{spd}$ | the gyrodistance in $\mathcal{S}_d^{++}$ under metrics $g$ |
| $\|\widetilde{\ominus}_{gr} \mathbf{U} \widetilde{\oplus}_{gr} \mathbf{V}\|^{gr}$ | the gyrodistance in $\widetilde{\mathcal{G}}(q, d)$ |
| $\left\|\left(\widetilde{\ominus}_{gr} \mathbf{U}_P \widetilde{\oplus}_{gr} \mathbf{U}_Q, \ominus_g \mathbf{S}_P \oplus_g \mathbf{S}_Q\right)\right\|_{psd}^g$ | the gyrodistance in $\widetilde{\mathcal{G}}(q, d) \times \mathcal{S}_d^{++}$ under metrics $g$ |
| $[\cdot, \cdot]$ | the matrix commutator |
| $\mathrm{expm}(\cdot), \mathrm{logm}(\cdot)$ | Matrix exponentiation and logarithm |
| $\mathscr{L}(\cdot), \mathscr{L}^{-1}(\cdot)$ | Cholesky decomposition and its inverse |
| $\mathbb{D}(\cdot)$ | A diagonal matrix with diagonal elements from a square matrix |
| $\lfloor \cdot \rfloor$ | The strictly lower triangular part of a square matrix |
| $\mathrm{Log}_{\mathbf{P}}^{gr}(\mathbf{Q})$ | Logarithmic map of $\mathbf{Q}$ at $\mathbf{P}$ in $\mathcal{G}(q, d)$ |
| $\mathcal{M}, \mathcal{N}$ | Matrix manifold |
| WFM | the weighted Fréchet mean |
| $\mathrm{hom}_{ai}(\cdot), \mathrm{hom}_{le}(\cdot), \mathrm{hom}_{lc}(\cdot)$ | the maps in $\mathcal{S}_d^{++}$ under AIM, LEM, and LCM satisfying gyro homomorphism |
| $\mathrm{hom}_{gr}(\cdot)$ | the maps in $\widetilde{\mathcal{G}}(q, d)$ satisfying gyro homomorphism |
| $\mathrm{hom}_{psd,g}(\cdot)$ | the maps in $\widetilde{\mathcal{G}}(q, d) \times \mathcal{S}_d^{++}$ under metrics $g$ satisfying gyro homomorphism |
| $\|\cdot\|_{\mathbf{F}}$ | The norm induced by the standard Frobenius inner product |
| $O(d)$ | The special orthogonal group |
| $\mathrm{Exp}_{\mathbf{P}}^{ai}(\mathbf{A})$ | Exponential map of $\mathbf{A}$ at $\mathbf{P}$ in $\mathcal{S}_d^{++}$ under AIM |
| $\mathrm{Log}_{\mathbf{P}}^{ai}(\mathbf{Q})$ | Logarithmic map of $\mathbf{Q}$ at $\mathbf{P}$ in $\mathcal{S}_d^{++}$ under AIM |
| $\mathrm{Exp}_{\mathbf{P}}^{gr}(\mathbf{W})$ | Exponential map of $\mathbf{W}$ at $\mathbf{P}$ in $\mathcal{G}(q, d)$ |
| $\widetilde{\mathrm{Exp}}_{\mathbf{X}}^{gr}(\mathbf{H})$ | Exponential map of $\mathbf{H}$ at $\mathbf{X}$ in $\widetilde{\mathcal{G}}(q, d)$ |
| $\widetilde{\mathrm{Log}}_{\mathbf{P}}^{gr}(\mathbf{Q})$ | Logarithmic map of $\mathbf{Q}$ at $\mathbf{P}$ in $\widetilde{\mathcal{G}}(q, d)$ |

Table A2: Summary of Abbreviations.

| Abbreviations | Explanation |
|---|---|
| SPD | Symmetric Positive Definite |
| SPSD | Symmetric Positive Semi-Definite |
| Homs | Homomorphisms |
| GryoAtt | Gyro Attention |
| EEG | Electroencephalography |
| LEM | Log-Euclidean Metric |
| LCM | Log-Cholesky Metric |
| AIM | Affine-Invariant Metric |
| WFM | Weighted Fréchet Mean |
| ONB | Orthonormal Basis |

## B IMPLEMENTATION DETAILS AND ADDITIONAL EXPERIMENTS

The Brain-computer Interface (BCI) enables direct interaction between the brain and external devices using electrical brain activity. Numerous applications in non-invasive BCI systems depend on effective modeling and information extraction from Electroencephalography (EEG) signals. EEG is a technique for measuring neural activity by high temporal resolution capturing the electric fields generated by the human scalp (Subha et al., 2010). Variations in rhythmic brain activity reflect cognitive processes (Pfurtscheller & Lopes, 1999), emotional states (Faller et al., 2019), and health conditions (Zhang et al., 2021). However, EEG signals exhibit a low signal-to-noise ratio (SNR) and low specificity, complicating meaningful information extraction (Johnson, 2006; Hine et al., 2017).

Table A3: GyroAtt-SPSD architectures across four datasets. The $q$ is the rank of the SPSD matrices.

| Block | MAMEM-SSVEP-II | BCI-ERN | BNCI2014001 | BNCI2015001 | Operation |
|---|---|---|---|---|---|
| Input data | $1 \times 8 \times 125$ | $1 \times 56 \times 160$ | $1 \times 22 \times 750$ | $1 \times 13 \times 768$ | |
| TempConv | $125 \times 1 \times 125$ | $22 \times 1 \times 160$ | $4 \times 22 \times 750$ | $5 \times 13 \times 768$ | Convolution |
| SpatConv | $21 \times 1 \times 126$ | $57 \times 1 \times 161$ | $43 \times 1 \times 750$ | $44 \times 1 \times 768$ | Convolution |
| Split & CovPool | $2 \times 21 \times 21$ | $3 \times 19 \times 19$ | $6 \times 43 \times 43$ | $3 \times 44 \times 44$ | Split + Covariance |
| SPDDSMBN | w/o | w/o | $6 \times 43 \times 43$ | $3 \times 44 \times 44$ | Domain Alignment Kobler et al. (2022) |
| SPSDCom | $2 \times (21 \times 9, 9 \times 9)$ | $3 \times (19 \times q, q \times q)$ | $6 \times (43 \times 18, 18 \times 18)$ | $3 \times (44 \times 18, 18 \times 18)$ | Alg. A3 |
| GyroAtt-SPSD | $2 \times (21 \times 9, 9 \times 9)$ | $3 \times (19 \times q, q \times q)$ | $6 \times (43 \times 18, 18 \times 18)$ | $3 \times (44 \times 18, 18 \times 18)$ | Alg. 1 |
| R2E | $2 \times (21 \times 21, 9 \times 9)$ | $3 \times (19 \times 19, q \times q)$ | $6 \times (43 \times 43, 18 \times 18)$ | $3 \times (44 \times 44, 18 \times 18)$ | $(\Phi(\cdot), \psi(\cdot))$ |
| Flat | $(882, 162)$ | $(1083, q^2)$ | $(11094, 1944)$ | $(5547, 972)$ | Vectorization |
| Classifier | 5 | 2 | 4 | 2 | FC + Softmax |

### B.1 DATASETS

**MAMEM-SSVEP-II.** This dataset includes EEG recordings from 11 subjects performing SSVEP tasks. Participants focused on one of five visual stimuli flickering at different frequencies for five seconds. Each subject completed five sessions, with five trials per stimulation frequency in each session. EEG signals were captured with 256 channels at a sampling rate of 250 Hz.

**BCI-ERN.** This dataset involves 26 subjects in a P300-based spelling task to measure ERN. EEG data were recorded from 56 electrodes following the extended 10-20 system at a sampling rate of 600 Hz. Each subject underwent five sessions: the first four with 60 trials each and the fifth with 100 trials. We used data from 16 subjects available in the initial competition release.

**BNCI2014001.** This dataset comprises EEG recordings from 9 subjects performing four motor imagery tasks: imagining movements of the left hand, right hand, both feet, and tongue. Each subject participated in two sessions on different days, each containing six runs. Each run included 48 trials—12 per class—totaling 288 trials per session.

**BNCI2015001.** EEG signals were recorded from electrodes centered around positions C3, Cz, and C4, according to the International 10-20 System. Data were collected using a g.GAMMAsys active electrode system with a g.USBamp amplifier, sampled at 512 Hz with a bandpass filter between 0.5 and 100 Hz and a notch filter at 50 Hz.

### B.2 EEG PREPROCESSING

For the BNCI2014001 and BNCI2015001 datasets, we followed the preprocessing steps described by Kobler et al. (2022). Using the Python packages moabb and mne, we resampled the EEG signals to 250/256 Hz, applied temporal filters to extract oscillatory activity in the 4–36 Hz range, and extracted short segments ($\leq 3$ seconds) associated with class labels.

For the MAMEM-SSVEP-II dataset, we adhered to the preprocessing protocol of Pan et al. (2022). The steps included: (1) band-pass filtering between 1–50 Hz; (2) selecting eight channels (PO7, PO3, PO, PO4, PO8, O1, Oz, and O2) located in the occipital area corresponding to the visual cortex; and (3) segmenting each trial into four 1-second segments from 1s to 5s after cue onset. This resulted in 500 trials of 1-second, 8-channel SSVEP signals per subject, with each input EEG segment comprising 125 time points.

For the BCI-ERN dataset, we followed the preprocessing procedure outlined by Pan et al. (2022). The steps involved: (1) downsampling the signals from 600 Hz to 128 Hz; (2) applying a band-pass filter between 1–40 Hz. After preprocessing, each trial consisted of 56 channels with 160 time points.

### B.3 ADDITIONAL IMPLEMENTATION DETAILS

Table A4: GyroAtt-SPD architectures across four datasets.

| Block | MAMEM-SSVEP-II | BCI-ERN | BNCI2014001 | BNCI2015001 | Operation |
|---|---|---|---|---|---|
| Input data | $1 \times 8 \times 125$ | $1 \times 56 \times 160$ | $1 \times 22 \times 750$ | $1 \times 13 \times 768$ | |
| TempConv | $125 \times 1 \times 125$ | $22 \times 1 \times 160$ | $4 \times 22 \times 750$ | $5 \times 13 \times 768$ | Convolution |
| SpatConv | $21 \times 1 \times 126$ | $57 \times 1 \times 161$ | $43 \times 1 \times 750$ | $44 \times 1 \times 768$ | Convolution |
| Split & CovPool | $2 \times 21 \times 21$ | $3 \times 19 \times 19$ | $6 \times 43 \times 43$ | $3 \times 44 \times 44$ | Split + Covariance |
| SPDDSMBN | w/o | w/o | $6 \times 43 \times 43$ | $3 \times 44 \times 44$ | Domain Alignment |
| GyroAtt-SPD | $2 \times 21 \times 21$ | $3 \times 19 \times 19$ | $6 \times 43 \times 43$ | $3 \times 44 \times 44$ | Alg. 1 |
| R2E | $2 \times 21 \times 21$ | $3 \times 19 \times 19$ | $6 \times 43 \times 43$ | $3 \times 44 \times 44$ | $\psi(\cdot)$ |
| Flat | 882 | 1083 | 11094 | 5547 | Vectorization |
| Classifier | 5 | 2 | 4 | 2 | FC + Softmax |

Table A5: GyroAtt-Gr Architectures across four datasets. The $q$ is the dimension of the linear subspaces.

| Block | MAMEM-SSVEP-II | BCI-ERN | BNCI2014001 | BNCI2015001 | Operation |
|---|---|---|---|---|---|
| Input data | $1 \times 8 \times 125$ | $1 \times 56 \times 160$ | $1 \times 22 \times 750$ | $1 \times 13 \times 768$ | |
| TempConv | $125 \times 1 \times 125$ | $22 \times 1 \times 160$ | $4 \times 22 \times 750$ | $5 \times 13 \times 768$ | Convolution |
| SpatConv | $21 \times 1 \times 126$ | $57 \times 1 \times 161$ | $43 \times 1 \times 750$ | $43 \times 1 \times 768$ | Convolution |
| Split & CovPool | $2 \times 21 \times 21$ | $3 \times 19 \times 19$ | $6 \times 43 \times 43$ | $3 \times 44 \times 44$ | Split + Covariance |
| SPDDSMBN | w/o | w/o | $6 \times 43 \times 43$ | $3 \times 44 \times 44$ | Domain Alignment |
| GrCom | $2 \times 21 \times 9$ | $3 \times 19 \times q$ | $6 \times 43 \times 18$ | $3 \times 44 \times 18$ | Alg. A3 |
| GyroAtt-Gr | $2 \times 21 \times 9$ | $3 \times 19 \times q$ | $6 \times 43 \times 18$ | $3 \times 44 \times 18$ | Alg. 1 |
| R2E | $2 \times 21 \times 21$ | $3 \times 19 \times 19$ | $6 \times 43 \times 43$ | $3 \times 44 \times 44$ | $\Phi(\cdot)$ |
| Flat | 882 | 1083 | 11094 | 5547 | Vectorization |
| Classifier | 5 | 2 | 4 | 2 | FC + Softmax |

Table A3 provides a summary of the specific network architectures of GyroAtt-SPSD across the four datasets. The network structures for GyroAtt-Gr (Tab. A5) and GyroAtt-SPD (Tab. A4) are identical to that of GyroAtt-SPSD. We just introduce GyroAtt-SPSD as an example.

For the MAMEM-SSVEP-II and BCI-ERN datasets, the initial convolutional block consists of a convolutional layer, followed by batch normalization and an ELU activation function. The subsequent convolutional block performs depthwise spatial convolution. A pointwise convolution, batch normalization, and another ELU activation follow this. In the MAMEM-SSVEP-II dataset, features are split into two non-overlapping segments, followed by covariance pooling. For the BCI-ERN dataset, the second convolutional block is repeated in two additional blocks. The outputs from these blocks are concatenated along the channel dimension. The data is then split along the channel dimension, and covariance pooling is applied, resulting in three covariance matrices.

For BNCI2014001 and BNCI2015001 datasets, the initial convolutional layer employs 4 or 5 filters with a kernel size of $(1, 25)$, performing temporal convolution while maintaining the same size through padding. The second convolutional layer applies spatial convolution with a kernel size of $(22, 1)$ to integrate information from different channels. The output sequences undergo temporal pyramid partitioning, dividing each sequence into $i$ equal segments at the $i$-th level (with levels set to 3 and 2, respectively). To address distribution shifts across subjects and runs, we incorporate subject- and run-specific batch normalization layers (Kobler et al., 2022).

The attention module designed in the gyrovector spaces is constituted by five operation layers, which are the Gyro homomorphism layer ($f_{\mathrm{hom}}$) used to generate $\mathbf{Q}_i$, $\mathbf{K}_i$, and $\mathbf{V}_i$ for each input data, the similarity measurement layer ($f_{\mathrm{sim}}$) for computing the correlation between $\mathbf{Q}_i$ and $\mathbf{K}_j$, the Softmax layer ($f_{\mathrm{smx}}$) used to normalize the obtained attention matrix along the row direction, the weighted Fréchet Mean layer ($f_{\mathrm{wFM}}$) for the implementation of weighted aggregation, and the power-based nonlinear activation layer ($f_{\mathrm{pac}}$) used to improve the representational capacity of GyroAtt module by introducing nonlinearity to the underlying metric space.

For classification, our GyroAtt-SPD model employs matrix power normalization following Wang et al. (2020) and Chen et al. (2024c). Specifically, we apply the transformation $\psi_\theta(\mathbf{P}) = \frac{1}{\theta}\mathbf{P}^\theta$ to the $i$-th output matrix $\mathbf{P} \in \mathcal{S}_d^{++}$, where $\theta > 0$. The coefficient $\frac{1}{\theta}$ stabilizes the gradient flow

during training and facilitates convergence. In GyroAtt-Gr, we transform elements $\mathbf{Y}_i \in \mathcal{G}(q,d)$ by applying a projection operator $\Phi(\mathbf{Y}_i) = \mathbf{Y}_i \mathbf{Y}_i^\top$ to map them into the corresponding flat space. In contrast, for GyroAtt-SPSD, we project $(\mathbf{U}_X^i, \mathbf{S}_X^i) \in \widetilde{\mathcal{G}}(q,d) \times \mathcal{S}_q^{++}$ onto their respective manifolds. In all three GyroAtt, the transformed matrices are vectorized, concatenated, and fed into a fully connected layer followed by a $\mathrm{Softmax}$ function.

## B.4 ABLATIONS ON THE GYROATT COMPONENTS

We conducted an ablation study to evaluate the contributions of the Gyro Homomorphism and nonlinear activation in GyroAtt. Specifically, we replaced these components in GyroAtt-SPD and GyroAtt-SPSD with equivalent layers from SPDNet and GrNet, such as Bimap, Frmap, and ReEig, to assess their impact on performance.

Table A6: Ablations of GyroAtt-SPD, Replacing Gyro Homomorphisms and Power Activations with SPDNet methods (The Bimap and ReEig layer). The best result under each geometry is highlighted in **bold**.

| Transformation | Activation | BNCI2014001 | | BNCI2015001 | |
|---|---|---|---|---|---|
| | | Inter-session | Inter-subject | Inter-session | Inter-subject |
| Bimap | Power | $74.0 \pm 6.5$ | $52.3 \pm 15.0$ | $85.2 \pm 7.2$ | $77.2 \pm 13.2$ |
| Homomorphisms | ReEig | $75.1 \pm 6.3$ | $52.6 \pm 14.2$ | $85.9 \pm 5.3$ | $76.4 \pm 12.8$ |
| Bimap | ReEig | $73.6 \pm 6.8$ | $52.2 \pm 15.2$ | $85.4 \pm 7.8$ | $76.8 \pm 13.0$ |
| Homomorphisms | Power | $75.4 \pm 7.4$ | $53.1 \pm 14.1$ | $86.2 \pm 4.5$ | $77.9 \pm 13.0$ |

Table A7: Ablations of GyroAtt-SPSD, Replacing Gyro Homomorphisms and Power Activations with SPDNet or GrNet methods (The Frmap and ReEig layers).

| Transformation | Activation | BNCI2014001 | | BNCI2015001 | |
|---|---|---|---|---|---|
| | | Inter-session | Inter-subject | Inter-session | Inter-subject |
| Frmap | Power | $68.9 \pm 6.9$ | $51.2 \pm 12.9$ | $82.3 \pm 6.2$ | $65.8 \pm 13.1$ |
| Homomorphisms | ReEig | $72.3 \pm 6.9$ | $49.6 \pm 13.3$ | $84.9 \pm 6.2$ | $74.1 \pm 12.3$ |
| Frmap | ReEig | $68.8 \pm 7.2$ | $50.7 \pm 13.8$ | $81.6 \pm 6.1$ | $72.9 \pm 13.3$ |
| Homomorphisms | Power | $72.9 \pm 6.2$ | $52.4 \pm 15.6$ | $85.3 \pm 5.3$ | $76.0 \pm 14.1$ |

**Implementation of component replacement on GyroAtt.** We replaced components in GyroAtt with their equivalents from MAtt and GDLNet to assess their contributions. Specifically, in GyroAtt-SPD, we replaced the Gyro Homomorphism $\mathrm{hom}(\cdot)$ with the Bimap layer and the matrix power-based nonlinear activation $\sigma(\cdot)$ with the ReEig layer. In GyroAtt-SPSD, we replaced $\mathrm{hom}(\cdot)$ with the Frmap layer and $\sigma(\cdot)$ with the ReEig layer. That is, we substituted $\mathrm{hom}_{gr}(\mathbf{U}_P)$ in $\mathrm{hom}_{psd,g}(\cdot)$ with the Frmap layer and replaced $(\mathbf{S}_{R_i})^p$ in $(\mathbf{U}_{R_i}, (\mathbf{S}_{R_i})^p)$ with the ReEig layer.

The BiMap (bilinear transformation) layer is defined as:

$$\mathbf{X}^{(l)} = \mathbf{W}^{(l)} \mathbf{X}^{(l-1)} \mathbf{W}^{(l)^\top}, \tag{A1}$$

where $\mathbf{X}^{(l)} \in \mathcal{S}_{d2}^{++}, \mathbf{X}^{(l-1)} \in \mathcal{S}_{d1}^{++}, \mathbf{W}^{(l)} \in \mathbb{R}^{d_2 \times d_1}$ with $d_1 > d_2$ is a semi-orthogonal matrix. For the parameter $\mathbf{W}^{(l)}$, we use the geoopt (Kochurov et al., 2020) package to optimize. The FrMap layer is defined as:

$$\mathbf{X}^{(l)} = \mathbf{W}^{(l)^\top} \mathbf{X}^{(l-1)}, \tag{A2}$$

where $\mathbf{X}^{(l)} \in \mathcal{G}(d_2, q)$, $\mathbf{X}^{(l-1)} \in \mathcal{G}(d_1, q)$, and $\mathbf{W}^{(l)} \in \mathbb{R}^{d_2 \times d_1}$ is a semi-orthogonal matrix with $d_1 > d_2$. We optimized $\mathbf{W}^{(l)}$ using Geoopt.

The ReEig (rectified eigenvalues activation) layer is defined as:

$$\mathbf{X}^l = \mathbf{U}^{(l)} \max(\mathbf{\Sigma}^{(l)}, \epsilon \mathbf{I}_d) \mathbf{U}^{(l)^\top}, \tag{A3}$$

with $\mathbf{X}^{l-1} = \mathbf{U}^{(l)} \mathbf{\Sigma}^{(l)} \mathbf{U}^{(l)^\top}$, where $\mathbf{\Sigma}^{(l)}$ contains the eigenvalues of $\mathbf{X}^{l-1}$, and $\epsilon \mathbf{I}_d$ is used to ensure numerical stability and set by 1e-4. Here, we set the dimensions of the Bimap layer to $21 \times 18$, $43 \times 20$, and $44 \times 20$ and the frmap layer to $21 \times 18$, $43 \times 30$, and $44 \times 30$ for the MAMEM-SSVEP-II, BNCI2014001, and BNCI2015001 datasets, respectively.

As shown in Tab. A6 and Tab. A7, replacing $\mathrm{hom}(\cdot)$ with the Bimap layer or $\sigma(\cdot)$ with the ReEig layer leads to significant performance degradation across the datasets. Similarly, for GyroAtt-SPSD, replacing $\mathrm{hom}(\cdot)$ with Frmap or $\sigma(\cdot)$ with ReEig degrades performance. This occurs because $\mathrm{hom}(\cdot)$ and $\sigma(\cdot)$ respect the gyro algebraic structure and underlying Riemannian geometry. The $\mathrm{hom}(\cdot)$ function, as a Gyro homomorphism, preserves the Gyro algebraic structure of $\oplus$ and $\otimes$, serving as a natural generalization of linear transformations in Euclidean spaces. In contrast, Bimap lacks these properties. Similarly, $\sigma(\cdot)$ introduces nonlinearity to SPD matrices and, more importantly, acts as an activation and deformation mechanism for the Riemannian metric, as discussed in Chen et al. (2024d). On the other hand, to some extent, ReEig is primarily a numerical activation method, ensuring only $\mathcal{S}_d^{++} \to \mathcal{S}_d^{++}$ without addressing these deeper structural and geometric considerations.

### B.5 ABLATIONS ON THE SIMILARITY CALCULATION IN GYROATT

In Euclidean space, attention mechanisms commonly use the inner product as similarity measures. Nguyen & Yang (2023); Nguyen et al. (2024) extends this concept by defining the inner product on SPD, SPSD, and Grassmannian manifolds. The specific formulations are detailed as follows:

Table A8: Ablations of GyroAtt, Replacing distance-based similarity to inner product-based similarity, where BNCI2014001 and BNCI2015001 datasets under inter-session settings.

| Methods | similarity | BNCI2014001 | BNCI2015001 | MAMEM-SSVEP-II |
|---|---|---|---|---|
| GyroAtt-SPD | inner product | $74.7 \pm 6.8$ | $85.6 \pm 5.4$ | $63.9 \pm 3.2$ |
| | geodesic distance | $75.4 \pm 7.4$ | $86.2 \pm 4.5$ | $66.3 \pm 2.2$ |
| GyroAtt-Gr | inner product | $72.4 \pm 7.3$ | $83.4 \pm 5.9$ | $65.7 \pm 3.1$ |
| | geodesic distance | $72.5 \pm 7.3$ | $85.0 \pm 7.7$ | $67.1 \pm 1.6$ |
| GyroAtt-SPSD | inner product | $71.6 \pm 6.3$ | $83.3 \pm 5.4$ | $65.0 \pm 2.6$ |
| | geodesic distance | $72.9 \pm 6.2$ | $85.3 \pm 5.3$ | $68.7 \pm 1.5$ |

For $\mathbf{P}, \mathbf{Q} \in \mathcal{S}_d^{++}$, the SPD inner product is given by (Nguyen & Yang, 2023):

$$\langle \mathbf{P}, \mathbf{Q} \rangle^g = \langle \mathrm{Log}_{\mathbf{I}_d}^g(\mathbf{P}), \mathrm{Log}_{\mathbf{I}_d}^g(\mathbf{Q}) \rangle_{\mathbf{I}_d}^g, \tag{A4}$$

For $\mathbf{U}, \mathbf{V} \in \widetilde{\mathcal{G}}(q, d)$, the inner product is given by:

$$\langle \mathbf{U}, \mathbf{V} \rangle^{gr} = \langle \widetilde{\mathrm{Log}}_{\widetilde{\mathbf{I}}_{d,q}}^{gr}(\mathbf{U}), \widetilde{\mathrm{Log}}_{\widetilde{\mathbf{I}}_{d,q}}^{gr}(\mathbf{V}) \rangle_{\widetilde{\mathbf{I}}_{d,q}}, \tag{A5}$$

For $(\mathbf{U}_P, \mathbf{S}_P), (\mathbf{U}_Q, \mathbf{S}_Q) \in \widetilde{\mathcal{G}}(q, d) \times \mathcal{S}_q^{++}$, the inner product is defined as:

$$\langle (\mathbf{U}_P, \mathbf{S}_P), (\mathbf{U}_Q, \mathbf{S}_Q) \rangle^{psd,g} = \lambda \langle \mathbf{U}_P \mathbf{U}_P^\top, \mathbf{U}_Q \mathbf{U}_Q^\top \rangle_{\widetilde{\mathbf{I}}_{d,q}}^{gr} + \langle \mathbf{S}_P, \mathbf{S}_Q \rangle_{\mathbf{I}_q}^g, \tag{A6}$$

We replaced the distance-based similarity computation in Eq. (8) with the inner product defined in follow and conducted ablation experiments on the MAMEM, BNCI2014001, and BNCI2015001 datasets under inter-session settings.

The results show that GyroAtt with inner product-based similarity generally performs worse than with geodesic distance-based similarity across most datasets. This is because the geodesic distance measures the shortest path between two points along the curved manifold surface. In contrast, the inner product has notable limitations. It operates in the tangent space, which provides only a linear approximation of the manifold around $\mathbf{I}_d$. Additionally, it depends on the $\mathbf{I}_d$, meaning the tangent space approximation is localized and may not accurately represent relationships between points farther from $vecI_d$. This restricts its ability to model global relationships on the manifold effectively.

### B.6 ABLATIONS ON THE MATRIX POWER NORMALIZATION

We conduct ablation experiments to assess the impact of the power normalization parameter $\theta$ on the performance of the proposed GyroAtt, as summarized in Tab. A9. For each gyro structure, we let the parameter $\theta$ vary within the set $\{0.25, 0.50, 0.75\}$. Among the SPD-based configurations, our GyroAttNet under SPD-AIM geometry achieves the highest inter-session accuracy on the BNCI2014001 dataset and the best inter-subject accuracy on the BNCI2015001 dataset at $p = 0.5$.

Table A9: Ablations of GyroAtt on matrix power normalization $\theta$ used in classification and Riemannian metrics. The best result under each geometry is highlighted in **bold**.

| Geometry | $\theta$ | BNCI2014001 | | BNCI2015001 | | MAMEM-SSVEP-II |
| | | Inter-session | Inter-subject | Inter-session | Inter-subject | |
|---|---|---|---|---|---|---|
| SPD-AIM | 0.25 | $74.9 \pm 6.9$ | $51.2 \pm 13.6$ | $86.1 \pm 7,3$ | $76.2 \pm 12.8$ | $61.9 \pm 2.5$ |
| | 0.50 | $\mathbf{75.4 \pm 7.1}$ | $\mathbf{53.1 \pm 14.8}$ | $\mathbf{86.2 \pm 4.5}$ | $\mathbf{77.9 \pm 13.0}$ | $66.2 \pm 2.8$ |
| | 0.75 | $75.0 \pm 8.1$ | $51.7 \pm 14.5$ | $86.0 \pm 6.5$ | $77.1 \pm 14.3$ | $\mathbf{66.3 \pm 2.2}$ |
| SPD-LEM | 0.25 | $75.2 \pm 6.7$ | $\mathbf{52.7 \pm 12.9}$ | $85.1 \pm 5.7$ | $\mathbf{76.9 \pm 14.5}$ | $60.7 \pm 2.4$ |
| | 0.50 | $\mathbf{75.3 \pm 6.5}$ | $51.4 \pm 14.1$ | $85.7 \pm 5.5$ | $76.6 \pm 13.7$ | $66.1 \pm 2.8$ |
| | 0.75 | $75.1 \pm 7.3$ | $52.3 \pm 13.3$ | $\mathbf{85.8 \pm 6.3}$ | $76.4 \pm 13.1$ | $\mathbf{66.2 \pm 2.5}$ |
| SPD-LCM | 0.25 | $\mathbf{74.2 \pm 7.5}$ | $52.1 \pm 14.5$ | $85.6 \pm 5.9$ | $77.3 \pm 13.4$ | $64.5 \pm 2.9$ |
| | 0.50 | $74.0 \pm 8.2$ | $\mathbf{52.7 \pm 13.6}$ | $85.9 \pm 6.7$ | $\mathbf{77.4 \pm 13.2}$ | $64.3 \pm 2.8$ |
| | 0.75 | $74.1 \pm 7.8$ | $52.0 \pm 14.7$ | $\mathbf{86.0 \pm 5.3}$ | $75.8 \pm 13.8$ | $\mathbf{65.1 \pm 2.5}$ |
| SPSD-AIM | 0.25 | $72.7 \pm 7.0$ | $51.2 \pm 15.8$ | $84.0 \pm 6.8$ | $\mathbf{75.5 \pm 13.8}$ | $66.3 \pm 2.9$ |
| | 0.50 | $\mathbf{72.9 \pm 6.2}$ | $\mathbf{52.4 \pm 15.6}$ | $\mathbf{84.5 \pm 6.6}$ | $74.2 \pm 15.2$ | $\mathbf{66.3 \pm 2.4}$ |
| | 0.75 | $72.7 \pm 6.7$ | $50.0 \pm 15.2$ | $84.4 \pm 4.9$ | $75.3 \pm 13.5$ | $65.7 \pm 2.7$ |
| SPSD-LEM | 0.25 | $\mathbf{72.8 \pm 7.1}$ | $\mathbf{50.7 \pm 13.9}$ | $\mathbf{85.3 \pm 5.3}$ | $\mathbf{76.0 \pm 14.1}$ | $\mathbf{66.6 \pm 2.6}$ |
| | 0.50 | $72.5 \pm 6.6$ | $50.6 \pm 14.2$ | $84.5 \pm 5.8$ | $75.1 \pm 12.9$ | $66.5 \pm 1.9$ |
| | 0.75 | $72.7 \pm 7.4$ | $49.5 \pm 12.9$ | $84.3 \pm 4.8$ | $74.7 \pm 14.3$ | $66.2 \pm 1.7$ |
| SPSD-LCM | 0.25 | $72.1 \pm 7.4$ | $49.9 \pm 13.1$ | $\mathbf{85.1 \pm 4.8}$ | $74.9 \pm 12.6$ | $67.6 \pm 2.1$ |
| | 0.50 | $\mathbf{72.9 \pm 6.7}$ | $48.4 \pm 13.3$ | $84.1 \pm 5.6$ | $74.4 \pm 13.7$ | $68.1 \pm 1.6$ |
| | 0.75 | $71.6 \pm 6.1$ | $\mathbf{50.1 \pm 12.8}$ | $84.1 \pm 5.7$ | $\mathbf{75.0 \pm 12.9}$ | $\mathbf{68.7 \pm 1.5}$ |

For the SPSD-based settings, SPSD-LEM geometry consistently performs well across multiple metrics, especially for the inter-session scenario in BNCI2015001, where it achieves a top accuracy of 85.3. It also can be noted that smaller or larger values of $p$ (*e.g.*, 0.25 or 0.75) tend to yield lower accuracy in most cases. In contrast, a moderate value of $p = 0.5$ appears to be more suitable for both SPD and SPSD geometries, as it could maintain a good normalization power. Besides, GyroAtt tends to be less sensitive to changes in $\theta$ across all experimental scenarios. In short, these results confirm the effectiveness of the introduced matrix power normalization in classification.

## C  RIEMANNIAN GEOMETRY OF GRASSMANNIAN MANIFOLDS

We now present the exponential and logarithmic maps, as well as the parallel translation under the ONB perspective, followed by the project perspective.

For the Grassmannian manifold $\widetilde{\mathcal{G}}(q, d)$ in the ONB perspective, the exponential map at $\mathbf{X} \in \widetilde{\mathcal{G}}(q, d)$ is defined as

$$\widetilde{\mathrm{Exp}}_{\mathbf{X}}^{gr}(\mathbf{H}) = \mathbf{X}\mathbf{V}\cos\mathbf{\Sigma} + \mathbf{U}\sin\mathbf{\Sigma}, \tag{A7}$$

where $\mathbf{H}$ is a tangent vector at $\mathbf{X}$, and $\mathbf{U}\mathbf{\Sigma}\mathbf{V}^\top$ is the thin singular value decomposition (SVD) of $\mathbf{H}$:

$$\mathbf{U}\mathbf{\Sigma}\mathbf{V}^\top = \mathrm{thinSVD}(\mathbf{H}). \tag{A8}$$

The logarithmic map, which is the inverse of the exponential map, is given by

$$\widetilde{\mathrm{Log}}_{\mathbf{X}}^{gr}(\mathbf{Y}) = \mathbf{U}\tan^{-1}\mathbf{\Sigma}\mathbf{V}^\top, \tag{A9}$$

where $\mathbf{X}, \mathbf{Y} \in \widetilde{\mathcal{G}}(q, d)$, and

$$\mathbf{U}\mathbf{\Sigma}\mathbf{V}^\top = \mathrm{thinSVD}\left((\mathbf{I} - \mathbf{X}\mathbf{X}^\top)\mathbf{Y}(\mathbf{X}^\top\mathbf{Y})^{-1}\right). \tag{A10}$$

As stated in Edelman et al. (1998, Theorem 2.4), let $\mathbf{H}$ and $\Delta$ be tangent vectors at point $\mathbf{Y}$ on the Grassmann manifold. The parallel transport of $\Delta$ along the geodesic in the direction $\dot{\mathbf{Y}}(0) = \mathbf{H}$ is given by

$$\tau\Delta(t) = \left((\mathbf{Y}\mathbf{V} \quad \mathbf{U})\begin{pmatrix} -\sin(\mathbf{\Sigma}t) \\ \cos(\mathbf{\Sigma}t) \end{pmatrix}\mathbf{U}^\top + (\mathbf{I} - \mathbf{U}\mathbf{U}^\top)\right)\Delta. \tag{A11}$$

Shifting to the projector perspective for the Grassmannian manifold $\mathcal{G}(q, d)$, let $\mathbf{P} \in \mathcal{G}(q, d)$ and $\Delta \in T_{\mathbf{P}}\mathcal{G}(q, d)$. The exponential map is defined as (Bendokat et al., 2024)

$$\operatorname{Exp}_{\mathbf{P}}^{gr}(\Delta) = \operatorname{expm}([\Delta, \mathbf{P}])\mathbf{P}\operatorname{expm}(-[\Delta, \mathbf{P}]). \tag{A12}$$

As shown by Sakai (1996), two points are in each other's cut locus if there exists more than one shortest geodesic connecting them. When the exponential map $\operatorname{Exp}_{\mathbf{P}}^{gr}$ is restricted to the injectivity domain $\operatorname{ID}_{\mathbf{P}}$, for any $\mathbf{F} \in \mathcal{G}(q, d) \backslash \operatorname{Cut}_{\mathbf{P}}$, there exists a unique tangent vector $\Delta \in \operatorname{ID}_{\mathbf{P}} \subset T_{\mathbf{P}}\mathcal{G}(q, d)$ such that $\operatorname{Exp}_{\mathbf{P}}^{gr}(\Delta) = \mathbf{F}$. For such a point $\mathbf{F}$, the logarithmic map is given by

$$\operatorname{Log}_{\mathbf{P}}^{gr}(\mathbf{Q}) = [\Omega, \mathbf{P}], \tag{A13}$$

where $\mathbf{P}, \mathbf{Q} \in \operatorname{Gr}_{n,p}$, and $\Omega$ is calculated as

$$\Omega = \frac{1}{2}\log\left((\mathbf{I}_n - 2\mathbf{Q})(\mathbf{I}_n - 2\mathbf{P})\right). \tag{A14}$$

# D    WEIGHTED FRÉCHET MEAN

## D.1    WEIGHTED FRÉCHET MEAN ON SPD MANIFOLDS

---
**Algorithm A1:** Karcher Flow Algorithm on the SPD Manifold under AIM

---
**Input:** A set of SPD matrices $\mathbf{X}_{1\ldots N} \in \mathcal{S}_d^{++}$
    A set of weights $w_{1\ldots N} > 0$ with $\sum_i w_i = 1$
    Number of iterations $K$
**Output:** The WFM $\mathbf{G}_k \in \mathcal{S}_d^{++}$
Initialize $\mathbf{G}_0 = \mathbf{I}$
**for** $k \leftarrow 1$ **to** $K$ **do**
    $\left| \quad \mathbf{G}_k \leftarrow \operatorname{Exp}_{\mathbf{G}_{k-1}}^{ai}\left(\sum_{i=1}^N w_i \operatorname{Log}_{\mathbf{G}_{k-1}}^{ai}(\mathbf{X}_i)\right) \right.$
**end**

---

**Affine-Invariant Metric.**    We begin by introducing the exponential and logarithmic maps under the affine-invariant metric (AIM), followed by the Karcher flow algorithm.

On the manifold $\mathcal{S}_d^{++}$ endowed with AIM, the exponential map at a point $\mathbf{P} \in \mathcal{S}_d^{++}$ is given by (Absil et al., 2004):

$$\operatorname{Exp}_{\mathbf{P}}^{ai}(\mathbf{A}) = \mathbf{P}^{\frac{1}{2}}\operatorname{expm}\left(\mathbf{P}^{-\frac{1}{2}}\mathbf{A}\mathbf{P}^{-\frac{1}{2}}\right)\mathbf{P}^{\frac{1}{2}}, \tag{A15}$$

where $\mathbf{A} \in T_{\mathbf{P}}\mathcal{S}_d^{++}$ is a tangent vector at $\mathbf{P}$. The logarithmic map, which is the inverse of the exponential map, is defined as

$$\operatorname{Log}_{\mathbf{P}}^{ai}(\mathbf{Q}) = \mathbf{P}^{\frac{1}{2}}\operatorname{logm}\left(\mathbf{P}^{-\frac{1}{2}}\mathbf{Q}\mathbf{P}^{-\frac{1}{2}}\right)\mathbf{P}^{\frac{1}{2}}, \tag{A16}$$

for any $\mathbf{Q} \in \mathcal{S}_d^{++}$.

As shown in Alg. A1, the Karcher flow algorithm computes the weighted Fréchet mean (WFM) on the SPD manifold through an iterative process. In each iteration, the data points are projected onto the tangent space at the current estimate $\mathbf{G}_{k-1}$ using the logarithmic map (Eq. (A16)), a weighted average is calculated in this tangent space, and the result is mapped back to the manifold using the exponential map (Eq. (A15)). This algorithm is guaranteed to converge on manifolds with non-positive curvatures, such as $\mathcal{S}_d^{++}$ (Karcher, 1977). We initialize $\mathbf{G}_0$ as the identity matrix $\mathbf{I}$ and set the number of iterations $K = 1$.

**Log-Euclidean Metric.**    Under the log-Euclidean metric (LEM), the WFM has a closed-form expression provided by Chen et al. (2024b):

$$\mathbf{G} = \operatorname{expm}\left(\sum_{i=1}^N w_i \operatorname{logm}(\mathbf{X}_i)\right), \tag{A17}$$

where $\mathbf{X}_{1\ldots N} \in \mathcal{S}_d^{++}$, $w_{1\ldots N} > 0$, and $\sum_i w_i = 1$.

**Log-Cholesky Metric.** Similarly, for the log-Cholesky metric (LCM), the WFM also admits a closed-form solution as shown by Chen et al. (2024b):

$$\mathbf{G} = \mathscr{L}^{-1}\left(\sum_{i=1}^{N} w_i \lfloor \mathscr{L}(\mathbf{X}_i) \rfloor + \prod_{i=1}^{N} \mathbb{D}(\mathscr{L}(\mathbf{X}_i))^{w_i}\right), \tag{A18}$$

where $\mathbf{X}_{1\ldots N} \in \mathcal{S}_d^{++}$, $w_{1\ldots N} > 0$, and $\sum_i w_i = 1$.

## D.2 Weighted Fréchet Mean on Grassmannian Manifolds

---

**Algorithm A2:** Karcher Flow Algorithm on the Grassmannian Manifold under ONB Perspective

---

**Input:** A set of Grassmannian points $\mathbf{X}_{1\ldots N} \in \widetilde{\mathcal{G}}(q, d)$
       A set of weights $w_{1\ldots N} > 0$ with $\sum_i w_i = 1$
       Number of iterations $K$
**Output:** The WFM $\mathbf{G} \in \widetilde{\mathcal{G}}(q, d)$
Initialize $\mathbf{G}_0 = \mathbf{X}_1$
**for** $k \leftarrow 1$ **to** $K$ **do**
    $\mathbf{G}_k \leftarrow \widetilde{\mathrm{Exp}}_{\mathbf{G}_{k-1}}^{gr}\left(\sum_{i=1}^{N} w_i \widetilde{\mathrm{Log}}_{\mathbf{G}_{k-1}}^{gr}(\mathbf{X}_i)\right)$
**end**

---

As shown in Alg. A1, the Karcher flow algorithm computes the WFM on the Grassmannian manifold through an iterative process. We initialize $\mathbf{G}_0$ as the identity matrix $\mathbf{X}_i$ and set the number of iterations $K = 1$.

## D.3 Weighted Fréchet Mean on SPSD Manifolds

As demonstrated by Bonnabel & Sepulchre (2010), the WFM for a batch of points $\mathbf{X}_{1,\ldots N} \in \mathcal{S}_{d,q}^+$ can be expressed as $(\mathrm{WFM}_{\mathrm{gr}}(\mathbf{U}_X^i), \mathrm{WFM}_{\mathrm{spd}}^g(\mathbf{S}_X^i))$. Here, $\mathrm{WFM}_{\mathrm{gr}}$ denotes the WFM on the Grassmannian manifold, while $\mathrm{WFM}_{\mathrm{spd}}^g(\cdot)$ represents the WFM on the SPD manifold under metric $g$. The matrices $\mathbf{U}_X^i$ and $\mathbf{S}_X^i$ correspond to the canonical representation of $\mathbf{X}_i$.

# E Canonical Representation in SPSD

---

**Algorithm A3:** Computation of Canonical Representation in SPSD manifold

---

**Input:** A batch of SPSD matrices $\mathbf{X}_{1\ldots N} \in \mathrm{S}_{n,q}^+$
      A constant $\gamma \in [0, 1]$
**Output:** A batch of Canonical Representation $(\mathbf{U}_X^i, \mathbf{S}_X^i)_{i=1,\ldots,N}$ of SPSD manifold
$\mathbf{U}^m \leftarrow \widetilde{\mathbf{I}}_{n,q}$;
$(\mathbf{U}_i, \boldsymbol{\Sigma}_i, \mathbf{V}_i)_{i=1,\ldots,N} \leftarrow \mathrm{SVD}((\mathbf{X}_i)_{i=1,\ldots,N})$
$(\mathbf{U}_i)_{i=1,\ldots,N} \leftarrow (\mathbf{U}_i[:,:q])_{i=1,\ldots,N}$;
**if** *training* **then**
    $\mathbf{U} \leftarrow \mathrm{GrMean}((\mathbf{U}_i)_{i=1,\ldots,N})$
    $\mathbf{U}^m \leftarrow \mathrm{GrGeodesic}(\mathbf{U}^m, \mathbf{U}, \gamma)$
**end**
**for** $i \leftarrow 1$ **to** $N$ **do**
    $(\mathbf{U}_i)^\top \mathbf{U}^m = \mathbf{Y}_i(\cos \boldsymbol{\Sigma}_i)\mathbf{V}_i^\top$
    $(\mathbf{U}_X^i, \mathbf{S}_X^i) \leftarrow (\mathbf{U}_i \mathbf{Y}_i, \mathbf{V}_i \mathbf{Y}_i^\top \mathbf{U}_i^\top \boldsymbol{\Sigma}_i \mathbf{U}_i \mathbf{Y}_i \mathbf{V}_i^\top)$
**end**

---

Nguyen et al. (2024) introduced a canonical representation of $\mathbf{P}$ in the structure space $\widetilde{\mathcal{G}}(q, d) \times \mathcal{S}_q^{++}$. As shown in Alg. A3, we follow this approach to derive the canonical representation of each point in $\mathcal{S}_{d,q}^+$. Canonical Representation of SPSD matrices is obtained in three steps. This first is to impose a decomposition on $\mathbf{X}_i$, *i.e.*, $\mathbf{X}_i \simeq \mathbf{U}_i \boldsymbol{\Sigma}_i \mathbf{U}_i^\top$, where $\mathbf{U}_i \in \widetilde{\mathcal{G}}(q, d)$ and $\boldsymbol{\Sigma}_i \in \mathcal{S}_q^{++}$. Then we use the mean of $\mathbf{U}_i)_{i=1,\ldots,N}$ as the common subspace, and rotated $(\mathbf{U}^i, \boldsymbol{\Sigma}^i)$ to the identified common

subspace, denoted as $(\mathbf{U}_X^i, \mathbf{S}_X^i)$. Here, $\mathrm{GrMean}((\mathbf{U}_i)_{i=1,\dots,N})$ computes the Fréchet mean of its arguments, as described in Alg. A2, with weights set to $w_{1,\dots,N} = \frac{1}{N}$. $\mathrm{GrGeodesic}(\mathbf{U}^m, \mathbf{U}, \gamma)$ computes a point on a geodesic (Eq. (A11)) from $\mathbf{U}^m$ to $\mathbf{U}$ at step $\gamma$ ($\gamma = 0.1$ in our experiments).

## F  OPTIMIZATION

We address the optimization of parameters that are SPD matrices by modeling them within the space of symmetric matrices and applying the exponential map to the identity matrix.

For any parameter $\mathbf{P} \in \widetilde{\mathcal{G}}(d, q)$, we parameterize it using a matrix $\mathbf{B} \in \mathbb{R}^{q, d-q}$ such that

$$\begin{bmatrix} 0 & \mathbf{B} \\ -\mathbf{B}^\top & 0 \end{bmatrix} = [\mathrm{Log}_{\mathbf{I}_{n,p}}^{gr}(\mathbf{PP}^\top), \mathbf{I}_{n,p}]. \tag{A19}$$

With this parameterization, the parameter $\mathbf{P}$ can be computed as

$$\mathbf{P} = \exp\left(\begin{bmatrix} 0 & \mathbf{B} \\ -\mathbf{B}^\top & 0 \end{bmatrix}\right) \widetilde{\mathbf{I}}_{n,p}.$$

To optimize parameters $\mathbf{O} \in SO(n)$, we start by generating parameter $\mathbf{A} \in \mathbb{R}^{n \times n}$, then compute its skew-symmetric matrix $\mathbf{S} = \mathbf{A} - \mathbf{A}^\top$. With this parameterization, the parameter $\mathbf{P}$ can be computed as

$$\mathbf{O} = (\mathbf{I} - \mathbf{S})(\mathbf{I} + \mathbf{S})^{-1}, \tag{A20}$$

This approach enables us to optimize all parameters within Euclidean spaces, eliminating the need to employ optimization techniques specific to Riemannian manifolds.

## G  PROOFS OF THE THEOREMS IN THE MAIN PAPER

*Proof of Thm. 5.1* . The $\oplus_{ai}$ and $\otimes_{ai}$ are defined by:

$$\mathbf{P} \oplus_{ai} \mathbf{Q} = \mathbf{P}^{\frac{1}{2}} \mathbf{Q} \mathbf{P}^{\frac{1}{2}}. \tag{A21}$$

$$t \otimes_{ai} \mathbf{P} = \mathbf{P}^t \tag{A22}$$

We begin by showing that $\mathrm{hom}_{ai}(\cdot)$ satisfies Eq. (5). let $\mathrm{hom}_{ai}(\mathbf{P}) = \mathbf{OPO}^\top$, with any $\mathbf{P}, \mathbf{Q} \in \mathcal{S}_d^{++}$, then we have

$$\begin{aligned} \mathrm{hom}_{ai}(\mathbf{P}) \oplus_{ai} \mathrm{hom}_{ai}(\mathbf{Q}) &\overset{(1)}{=} \left(\mathbf{OPO}^\top\right)^{\frac{1}{2}} \mathbf{OQO}^\top \left(\mathbf{OPO}^\top\right)^{\frac{1}{2}} \\ &\overset{(2)}{=} \mathbf{OP}^{\frac{1}{2}} \mathbf{O}^\top \mathbf{OQO}^\top \mathbf{OP}^{\frac{1}{2}} \mathbf{O}^\top \\ &= \mathbf{OP}^{\frac{1}{2}} \mathbf{QP}^{\frac{1}{2}} \mathbf{O}^\top \\ &= \mathrm{hom}_{ai}(\mathbf{P} \oplus_{ai} \mathbf{Q}). \end{aligned} \tag{A23}$$

The derivation of Eq. (A23) follows.

(1) follow from Eqs. (11) and (A21).

(2) follows from the fact that $\mathbf{P}$ is an SPD matrix and $\mathbf{O}$ is an orthogonal matrix.

Now, we proof that $\mathrm{hom}_{ai}(\cdot)$ satisfies Eq. (6). For the $\otimes_{ai}$, we have

$$\begin{aligned} t \otimes_{ai} \mathrm{hom}_{ai}(\mathbf{P}) &\overset{(1)}{=} \left(\mathbf{OPO}^\top\right)^t \\ &\overset{(2)}{=} \mathbf{OP}^t \mathbf{O}^\top \\ &= \mathrm{hom}_{ai}(t \otimes_{ai} \mathbf{P}). \end{aligned} \tag{A24}$$

The derivation of Eq. (A24) follows.

(1) follow from Eqs. (11) and (A22).

(2) follows from the fact that $\mathbf{P}$ is an SPD matrix and $\mathbf{O}$ is an orthogonal matrix.  $\square$

*Proof of Thm. 5.2* . The $\oplus_{le}$ and $\otimes_{le}$ are defined by:

$$\mathbf{P} \oplus_{le} \mathbf{Q} = \mathrm{expm}(\mathrm{logm}(\mathbf{P}) + \mathrm{logm}(\mathbf{Q})), \tag{A25}$$

$$t \otimes_{le} \mathbf{P} = \mathbf{P}^t \tag{A26}$$

We begin by showing that $\mathrm{hom}_{le}(\cdot)$ satisfies Eq. (5). For the $\oplus_{le}$, with any $\mathbf{P}, \mathbf{Q} \in \mathcal{S}_d^{++}$, we have

$$\begin{aligned}
\mathrm{hom}_{le}(\mathbf{P}) \oplus_{le} \mathrm{hom}_{le}(\mathbf{Q}) &\overset{(1)}{=} \mathrm{expm}\left(\mathbf{M}\,\mathrm{logm}\,(\mathbf{P})\,\mathbf{M}^\top + \mathbf{M}\,\mathrm{logm}\,(\mathbf{Q})\,\mathbf{M}^\top\right) \\
&= \mathrm{expm}\left(\mathbf{M}\,(\mathrm{logm}\,(\mathbf{P}) + \mathrm{logm}\,(\mathbf{Q}))\,\mathbf{M}^\top\right) \\
&= \mathrm{hom}_{le}(\mathbf{P} \oplus_{le} \mathbf{Q}).
\end{aligned} \tag{A27}$$

The derivation of Eq. (A27) follows.

(1) follow from Eqs. (12) and (A25).

For $\otimes_{le}$, we have

$$\begin{aligned}
t \otimes_{le} \mathrm{hom}_{le}(\mathbf{P}) &\overset{(1)}{=} \left(\mathrm{expm}\left(\mathbf{M}\,\mathrm{logm}\,(\mathbf{P})\,\mathbf{M}^\top\right)\right)^t \\
&\overset{(2)}{=} \mathrm{expm}\left(t\mathbf{M}\,\mathrm{logm}\,(\mathbf{P})\,\mathbf{M}^\top\right) \\
&= \mathrm{hom}_{le}(t \otimes_{le} \mathbf{P}).
\end{aligned} \tag{A28}$$

$\square$

*Proof of Cor. 5.3* . For the $\oplus_{le}$, with any $\mathbf{P}, \mathbf{Q} \in \mathcal{S}_d^{++}$, $\mathbf{O} \in \mathrm{O}(d)$ we have

$$\begin{aligned}
\mathrm{hom}_{le}(\mathbf{P}) \oplus_{le} \mathrm{hom}_{le}(\mathbf{Q}) &\overset{(1)}{=} \mathrm{expm}\left(\mathbf{O}\,(\mathrm{logm}\,(\mathbf{P}) + \mathrm{logm}\,(\mathbf{Q}))\,\mathbf{O}^\top\right) \\
&\overset{(2)}{=} \mathbf{O}\,\mathrm{expm}\left((\mathrm{logm}\,(\mathbf{P}) + \mathrm{logm}\,(\mathbf{Q}))\right)\mathbf{O}^\top \\
&= \mathrm{hom}_{le}(\mathbf{P} \oplus_{ai} \mathbf{Q}).
\end{aligned} \tag{A29}$$

The derivation of Eq. (A29) follows.

(1) follow from Eqs. (A25) and (A27).

(2) follows from the fact that $\mathbf{P}$ is an SPD matrix and $\mathbf{O}$ is an orthogonal matrix.

For the $\otimes_{le}$, we have

$$\begin{aligned}
t \otimes_{le} \mathrm{hom}_{le}(\mathbf{P}) &\overset{(1)}{=} \mathrm{expm}\left(t\mathbf{O}\,\mathrm{logm}\,(\mathbf{P})\,\mathbf{O}^\top\right) \\
&\overset{(2)}{=} \mathbf{O}\,\mathrm{expm}\left(t\,\mathrm{logm}\,(\mathbf{P})\right)\mathbf{O}^\top \\
&= \mathrm{hom}_{le}(t \otimes_{le} \mathbf{P}).
\end{aligned} \tag{A30}$$

The derivation of Eq. (A30) follows.

(1) follow from Eqs. (A26) and (A28).

(2) follows from the fact that $\mathbf{P}$ is an SPD matrix and $\mathbf{O}$ is an orthogonal matrix. $\square$

*Proof of Thm. 5.4* . The $\oplus_{lc}$ and $\otimes_{lc}$ are defined by:

$$t \otimes_{lc} \mathbf{P} = \mathscr{L}^{-1}\left(t\lfloor\mathscr{L}(\mathbf{P})\rfloor + \mathbb{D}(\mathscr{L}(\mathbf{P}))^t\right), \tag{A31}$$

$$\mathbf{P} \oplus_{lc} \mathbf{Q} = \mathscr{L}^{-1}\left(\lfloor\mathscr{L}(\mathbf{P})\rfloor + \lfloor\mathscr{L}(\mathbf{Q})\rfloor + \mathbb{D}(\mathscr{L}(\mathbf{P}))\mathbb{D}(\mathscr{L}(\mathbf{Q}))\right). \tag{A32}$$

We begin by showing that $\mathrm{hom}_{lc}(\cdot)$ satisfies Eq. (5). With any $\mathbf{P}, \mathbf{Q} \in \mathcal{S}_d^{++}$, for $\oplus_{lc}$, we can rewrite $\oplus_{lc}$ and $\mathrm{hom}_{lc}$ as

$$\mathbf{P} \oplus_{lc} \mathbf{Q} = \mathscr{L}^{-1}\left(\exp\mathbb{D}\left(\log\mathbb{D}\left(\mathscr{L}(\mathbf{P})\right) + \log\mathbb{D}\left(\mathscr{L}(\mathbf{Q})\right)\right)\right), \tag{A33}$$

$$\mathrm{hom}_{lc}(\mathbf{P}) = \mathscr{L}^{-1}\left(\exp\mathbb{D}\left(L(\mathbf{P})\right)\right), \tag{A34}$$

where $L(\cdot)$ is given by Eq. (15), $\log\mathbb{D}\,(\mathbf{F})$ and $\exp\mathbb{D}\,(\mathbf{F})$ are given by

$$\log\mathbb{D}\,(\mathbf{F}) = \lfloor\mathbf{F}\rfloor + \mathrm{logm}(\mathbb{D}(\mathbf{F})), \tag{A35}$$

$$\exp\mathbb{D}\,(\mathbf{F}) = \lfloor\mathbf{F}\rfloor + \mathrm{expm}(\mathbb{D}(\mathbf{F})), \tag{A36}$$

Then we have

$$\hom_{lc}(\mathbf{P}) \oplus_{lc} \hom_{lc}(\mathbf{Q}) \overset{(1)}{=} \mathscr{L}^{-1}\left(\exp \mathbb{D}\left(L(\mathbf{P}) + L(\mathbf{Q})\right)\right)$$
$$\overset{(2)}{=} \mathscr{L}^{-1}\left(\exp \mathbb{D}\left(L(\mathbf{P} + \mathbf{Q})\right)\right) \tag{A37}$$
$$= \hom_{lc}(\mathbf{P} \oplus_{lc} \mathbf{Q})$$

The derivation of Eq. (A37) follows.

(1) follow from Eqs. (14) and (A32).

(2) follow from the properties of $L(\cdot)$. $\qquad\square$

*Proof of Thm. 5.5* . The $\widetilde{\oplus}_{gr}$ and $\widetilde{\otimes}_{gr}$ are defined by:

$$\mathbf{U}\widetilde{\oplus}_{gr}\mathbf{V} = \expm([\text{Log}^{gr}_{\mathbf{I}_{d,q}}(\mathbf{U}\mathbf{U}^\top), \mathbf{I}_{d,q}])\mathbf{V}, \tag{A38}$$

$$t\widetilde{\otimes}_{gr}\mathbf{U} = \expm\left(\left[t\,\text{Log}^{gr}_{\mathbf{I}_{n,q}}, \mathbf{I}_{d,q}\right]\right)\mathbf{I}_{d,q} \tag{A39}$$

we begin by showing that $\hom_{gr}(\cdot)$ satisfies Eq. (5). For any $\mathbf{U}, \mathbf{V} \in \mathcal{G}(q, d)$, we have

$$\hom_{gr}(\mathbf{U})\widetilde{\oplus}_{gr}\hom_{gr}(\mathbf{V}) \overset{(1)}{=} \expm([\text{Log}^{gr}_{\mathbf{I}_{n,q}}(\mathbf{O}\mathbf{U}\mathbf{U}^\top\mathbf{O}^\top), \mathbf{I}_{n,q}])\mathbf{O}\mathbf{V}$$
$$\overset{(2)}{=} \expm([\mathbf{O}\,\text{Log}^{gr}_{\mathbf{I}_{n,q}}(\mathbf{U}\mathbf{U}^\top)\mathbf{O}^\top, \mathbf{O}\mathbf{I}_{n,q}\mathbf{O}^\top])\mathbf{O}\mathbf{V}$$
$$= \expm(\mathbf{O}[\text{Log}^{gr}_{\mathbf{I}_{n,q}}(\mathbf{U}\mathbf{U}^\top), \mathbf{I}_{n,q}]\mathbf{O}^\top)\mathbf{O}\mathbf{V}$$
$$\overset{(3)}{=} \mathbf{O}\expm([\text{Log}^{gr}_{\mathbf{I}_{n,q}}(\mathbf{U}\mathbf{U}^\top), \mathbf{I}_{n,q}])\mathbf{O}^\top\mathbf{O}\mathbf{V} \tag{A40}$$
$$= \mathbf{O}\expm([\text{Log}^{gr}_{\mathbf{I}_{n,q}}(\mathbf{U}\mathbf{U}^\top), \mathbf{I}_{n,q}])\mathbf{V}$$
$$= \hom_{gr}(\mathbf{U}\widetilde{\oplus}_{gr}\mathbf{V}).$$

The derivation of Eq. (A40) follows.

(1) follow from Eqs. (16) and (A38).

(2) follows from the fact that $\text{Log}^{gr}_{\mathbf{O}\mathbf{I}_{n,q}\mathbf{O}^\top}(\mathbf{O}\mathbf{U}\mathbf{U}^\top\mathbf{O}^\top) = \mathbf{O}\,\text{Log}^{gr}_{\mathbf{I}_{n,q}}(\mathbf{U}\mathbf{U}^\top)\mathbf{O}^\top$, and for $\mathbf{O} = \begin{bmatrix}\mathbf{O}_q & 0 \\ 0 & \mathbf{O}_{d-q}\end{bmatrix}, \mathbf{O}\mathbf{I}_{n,q}\mathbf{O}^\top = \mathbf{I}_{n,q}$.

(3) follows from the fact that $\mathbf{O}$ is an orthogonal matrix.

Now, we proof that $\hom_{gr}(\cdot)$ satisfies Eq. (6). The differential homomorphism $\Phi : \widetilde{\mathcal{G}}(q, d) \to \mathcal{G}(q, d), \mathbf{U} \to \mathbf{U}\mathbf{U}^\top$ exists between $\widetilde{\mathcal{G}}(q, d)$ and $\mathcal{G}(q, d)$, and $\widetilde{\otimes}_{gr}$ is derived from $\otimes_{gr}$ via this differential homomorphism. Thus, to prove that $\widetilde{\otimes}_{gr}$ satisfies Eq. (6), it suffices to show that $\otimes_{gr}$ satisfies Eq. (6). The $\otimes_{gr}$ is defined by:

$$t \otimes_{gr} \mathbf{U} = \expm\left(\left[t\bar{\mathbf{U}}, \mathbf{I}_{d,q}\right]\right)\mathbf{I}_{d,q}\expm\left(\left[-\bar{t}\mathbf{U}, \mathbf{I}_{d,q}\right]\right) \tag{A41}$$

For $\otimes_{gr}$, we have

$$t \otimes_{gr} \hom_{gr}(\mathbf{U}\mathbf{U}^\top) = (t\widetilde{\otimes}_{gr}\hom_{gr}(\mathbf{U}))(t\widetilde{\otimes}_{gr}\hom_{gr}(\mathbf{U}))^\top$$
$$\overset{(1)}{=} \expm(t[\text{Log}^{gr}_{\mathbf{I}_{n,q}}(\mathbf{O}\mathbf{U}\mathbf{U}^\top\mathbf{O}^\top), \mathbf{I}_{n,q}])\mathbf{I}_{n,q}\expm(t[\text{Log}^{gr}_{\mathbf{I}_{n,q}}(\mathbf{O}\mathbf{U}\mathbf{U}^\top\mathbf{O}^\top), \mathbf{I}_{n,q}])$$
$$\overset{(2)}{=} \mathbf{O}\expm([\text{Log}^{gr}_{\mathbf{I}_{n,q}}(\mathbf{U}\mathbf{U}^\top), \mathbf{I}_{n,q}])\mathbf{O}^\top\mathbf{I}_{n,q}\mathbf{O}\expm([\text{Log}^{gr}_{\mathbf{I}_{n,q}}(\mathbf{U}\mathbf{U}^\top), \mathbf{I}_{n,q}])\mathbf{O}^\top$$
$$= \mathbf{O}\expm([\text{Log}^{gr}_{\mathbf{I}_{n,q}}(\mathbf{U}\mathbf{U}^\top), \mathbf{I}_{n,q}])\mathbf{I}_{n,q}\expm([\text{Log}^{gr}_{\mathbf{I}_{n,q}}(\mathbf{U}\mathbf{U}^\top), \mathbf{I}_{n,q}])\mathbf{O}^\top$$
$$= \hom_{gr}(t \otimes_{gr} \mathbf{U}\mathbf{U}^\top).$$

$$\tag{A42}$$

Since $\otimes_{gr}$ satisfies Eq. (6), we can proof $\widetilde{\otimes}_{gr}$ satisfies Eq. (6). $\qquad\square$

*Proof of Thm. 5.6* . The $\widetilde{\oplus}_{psd,g}$ and $\otimes_{psd,g}$ are defined by:

$$(\mathbf{U}_P, \mathbf{S}_P) \oplus_{psd,g} (\mathbf{U}_Q, \mathbf{S}_Q) = (\mathbf{U}_P \widetilde{\oplus}_{gr} \mathbf{U}_Q, \mathbf{S}_P \oplus_g \mathbf{S}_Q), \tag{A43}$$

$$t \otimes_{psd,g} (\mathbf{U}_P, \mathbf{S}_P) = (t \widetilde{\otimes}_{gr} \mathbf{U}_P, t \otimes_g \mathbf{S}_P) \tag{A44}$$

we begin by showing that $\hom_{psd,g}$ satisfies Eq. (5). As shown in Eq. (17) For any $(\mathbf{U}_P, \mathbf{S}_P), (\mathbf{U}_Q, \mathbf{S}_Q) \in \widetilde{\mathcal{G}}(q,d) \times \mathcal{S}_q^{++}$, we have:

$$
\begin{aligned}
\hom_{psd,g}((\mathbf{U}_P, \mathbf{S}_P) \oplus_{psd,g} (\mathbf{U}_Q, \mathbf{S}_Q)) &\overset{(1)}{=} \hom_{psd,g}(\mathbf{U}_P \widetilde{\oplus}_{gr} \mathbf{U}_Q, \mathbf{S}_P \oplus_g \mathbf{S}_Q) \\
&= (\hom_{gr}(\mathbf{U}_P \widetilde{\oplus}_{gr} \mathbf{U}_Q), \hom_g(\mathbf{S}_P \oplus_g \mathbf{S}_Q)) \\
&\overset{(2)}{=} (\hom_{gr}(\mathbf{U}_P) \widetilde{\oplus}_{gr} \hom_{gr}(\mathbf{U}_Q), \hom_g(\mathbf{S}_P) \oplus_g \hom_g(\mathbf{S}_Q)) \\
&\overset{(3)}{=} (\hom_{gr}(\mathbf{U}_P), \hom_g(\mathbf{S}_P)) \oplus_{psd,g} (\hom_{gr}(\mathbf{U}_Q), \hom_g(\mathbf{S}_Q)) \\
&= \hom_{psd,g}(\mathbf{U}_P, \mathbf{S}_P) \oplus_{psd,g} \hom_{psd,g}(\mathbf{U}_Q, \mathbf{S}_Q).
\end{aligned}
\tag{A45}
$$

The derivation of Eq. (A45) follows.

(1) follow from Eqs. (17) and (A43).

(2) and (3) follow from the fact that $\hom_{gr}$ and $\hom_g$ are gyro homomorphisms.

For scalar multiplication, we have:

$$
\begin{aligned}
\hom_{psd,g}(t \otimes_{psd,g} (\mathbf{U}_P, \mathbf{S}_P)) &\overset{(1)}{=} \hom_{psd,g}(t \widetilde{\otimes}_{gr} \mathbf{U}_P, t \otimes_g \mathbf{S}_P) \\
&= (\hom_{gr}(t \widetilde{\otimes}_{gr} \mathbf{U}_P), \hom_g(t \otimes_g \mathbf{S}_P)) \\
&\overset{(2)}{=} (t \widetilde{\otimes}_{gr} \hom_{gr}(\mathbf{U}_P), t \otimes_g \hom_g(\mathbf{S}_P)) \\
&\overset{(3)}{=} t \otimes_{psd,g} (\hom_{gr}(\mathbf{U}_P), \hom_g(\mathbf{S}_P)) \\
&= t \otimes_{psd,g} \hom_{psd,g}(\mathbf{U}_P, \mathbf{S}_P).
\end{aligned}
\tag{A46}
$$

The derivation of Eq. (A46) follows.

(1) follow from Eqs. (17) and (A44).

(2) and (3) follow from the fact that $\hom_{gr}(\cdot)$ and $\hom_g(\cdot)$ are gyro homomorphisms.

$\square$

