# OpenReview forum: "GyroAtt: A Gyro Attention Framework for Matrix Manifolds"
_ICLR.cc/2025/Conference — Submitted to ICLR 2025_

### Official Review · Reviewer_nEzP · 2024-10-31

**Soundness:** 3
**Presentation:** 2
**Contribution:** 2
**Rating:** 6
**Confidence:** 4

**Summary:**

This article presents a framework that unifies attention mechanisms on different manifolds such as the SPD manifold using gyrovector spaces. The main operations of the attention mechanism are focused on feature transformation, attention calculation and aggregation. The key contribution is in the feature transformation part, where they introduce gyro homomorphisms. Relationships between the gyrovector space with other manifolds were implemented. They showed promising results on four EEG datasets.

**Strengths:**

- The key contribution is the introduction of gyro homomorphisms and its application to different manifolds and metrics. Their work specifically defines seven homomorphisms and their proofs were provided.
- Good results on the experiments.

**Weaknesses:**

- Details on the model structure are not clear.

**Questions:**

- The paper focuses on the attention mechanism, but the overall model structure used in the experiments remains unclear. For example, when the network operates on the SPD (GyroAtt-SPD), the attention mechanism appears identical to that in the Matt [1] network. It would be helpful if the authors clarify the reason for performance differences observed in the experiments between these two networks.
- One clear difference is in the activation function. Are there any other similar ones? It would be helpful if the authors include a comparison between their GyroAtt network and other non-Euclidean, attention-based networks used in the experiments. Such comparisons would greatly assist readers in better understanding of the presented results.
- Table 6 shows the model's performance across different power parameters (a parameter of the activation function). To better understand the contribution of the activation function itself, could the authors add a row for each experiment showing results without any activation function?
- At the standard Euclidean attention mechanism, a single head is barely used, as the empirical contribution of using multi-head is well known. Also, can the proposed attention mechanism be easily extended to multi-head configurations in different geometries?

[1 ] Yue-Ting Pan, Jing-Lun Chou, and Chun-Shu Wei. MAtt: A manifold attention network for EEG decoding. In NeurIPS, pp.31116–31129, 2022.

---

> ### Author Response · Authors · 2024-11-24
> **Response to Reviewer nEzP**
>
> $\textcolor{red}{\rm{Rebuttal}}$
>
> We thank Reviewer $\textcolor{purple}{nEzp (R4)}$ for the careful review and the suggestive comments. Below, we address the comments in detail. 😄
> ***
>
> **1. Model structure.**
> For detailed GyroAtt model structure, please refer to the **CQ#1** in Common Response.
>
>
> **2. The differences between GyroAtt-SPD and MAtt.**
>
> - **Applicability to Multiple Metrics:** GyroAtt-SPD implements attention mechanisms under three different metrics: Affine-Invariant Metric (AIM), Log-Euclidean Metric (LEM), and Log-Cholesky Metric (LCM). In contrast, MAtt [a] is based solely on the LEM. This broader applicability allows GyroAtt-SPD to be more versatile and adaptable to various manifold structures.
> - **Mathematical Principles of the Transformation Layers:** We utilize transformation layers called **Gyro Homomorphisms**, which preserve the algebraic structure of gyro spaces, including gyro addition and gyro scalar multiplication. These algebraic structures are derived from Riemannian operators on manifolds, ensuring that our Gyro Homomorphisms respect Riemannian geometry. In contrast, MAtt uses the **BiMap** layer, which is a numerical method that does not fully respect Riemannian geometry.
> - **Activation Function:** While both models use different activation functions, our method introduces a nonlinear activation function based on matrix power, which actually plays the role of nonlinear regularization of the metric space (enabling transition between different Riemannian metrics, e.g., ). Therefore, the model's ability to adapt to complex data distributions will be enhanced.
>
>
> **3. Comparisons with other non-Euclidean attention-based networks**
>
> Please refer to the common response **CQ#4**.
>
>
> **4. Contribution of the activation function**
>
>
> To further demonstrate the effectiveness of the introduced power-based activation function, we have made experiments on the BNCI2014001, BNCI2015001, and MAMEM-SSVEP-II datasets, analyzing the changes in classification accuracy for GyroAtt-SPD and GyroAtt-SPSD with and without this activation function. Tab. A illustrates the best results under each geometry, in which 'w/' and 'w/o' denote the inclusion and exclusion of this activation function, respextively. As shown, incorporating the power-based activation consistently leads to accuracy gains across all datasets and geometries, highlighting its significance in improving the models' representational capacity. We argue that the fundamental reasons lie in two aspects: 1) the power-based activation enhances the nonlinearity of GyroAtt, enabling it to better capture complex patterns; 2) regularizing the metric space could align the learned feature manifold more closely with the space assumed by the classifier, reducing geometric distortion when embedding manifold-valued data into Euclidean space. As a result, the learning ability of the classifier will be boosted. For additional details, please refer to the revised Tab. 6 in the manuscript.
>
>
> Table A: Ablations of GyroAtt on Riemannian metrics and matrix power activation $p$. The best result under each geometry is highlighted in **bold**.
> |Geometry|$p$|BNCI2014001 Inter-session|BNCI2014001 Inter-subject|BNCI2015001 Inter-session|BNCI2015001 Inter-subject|MAMEM-SSVEP-II|
> |-|-|-|-|-|-|-|
> |SPD-AIM|w/o|74.8±6.7|51.2±15.7|85.5±5.0|75.4±12.9|64.3±2.4|
> |SPD-AIM|w/|**75.4±7.1**|**53.1±14.8**|**86.2±4.5**|**77.9±13.0**|**66.3±2.2**|
> |SPD-LEM|w/o|74.9±7.3|51.7±15.8|85.2±5.2|75.3±12.3|65.6±2.3|
> |SPD-LEM|w/|**75.3±6.5**|**52.3±14.1**|**85.7±5.5**|**76.6±13.7**|**66.2±2.5**|
> |SPD-LCM|w/o|73.2±6.7|51.9±14.8|85.3±7.2|76.2±13.3|64.0±2.8|
> |SPD-LCM|w/|**74.2±7.8**|**52.7±13.6**|**86.0±6.8**|**77.4±13.2**|**65.1±2.5**|
> |SPSD-AIM|w/o|72.2±7.2|49.2±13.7|84.1±7.2|73.6±14.3|65.8±2.6|
> |SPSD-AIM|w/|**72.9±7.1**|**52.4±15.6**|**84.7±6.6**|**75.5±13.8**|**66.5±2.9**|
> |SPSD-LEM|w/o|72.1±6.7|49.8±12.9|83.9±5.1|74.2±14.4|66.2±1.9|
> |SPSD-LEM|w/|**72.8±6.9**|**50.5±13.2**|**85.3±5.3**|**76.0±14.1**|**66.5±2.3**|
> |SPSD-LCM|w/o|72.3±7.3|49.5±12.0|84.8±6.1|74.4±13.2|66.5±2.4|
> |SPSD-LCM|w/|**72.9±6.7**|**51.7±13.1**|**85.1±4.8**|**74.9±12.6**|**68.7±1.5**|
>
>
> **5. Extension to multi-head configurations**
>
> Our proposed attention mechanism can indeed be extended to multi-head configurations on the SPD manifold. Given an input data sample of size $N \times C \times D \times D$, we can partition it into $h$ heads, resulting in a reshaped data sample of size $N \times h \times \frac{C}{h} \times D \times D$. The multi-head attention mechanism is then computed independently for each head, and the outputs are concatenated. This approach allows the model to capture a richer set of relationships and can be directly applied to different geometries.
>
> **References**
>
> > [a] MAtt: A Manifold Attention Network for EEG Decoding. In NeurIPS 2022.

---

> > ### Comment · Reviewer_nEzP · 2024-11-26
> > **Feedback**
> >
> > I have no further comments.

---

> ### Author Response · Authors · 2024-11-26
>
> Thank you for your instant reply and suggestive comments! 😄😄

---

> ### Author Response · Authors · 2024-11-28
> **Additional Experiments on Multi-Head Attention**
>
> **Multi-Head Attention**
>
> Table B: Performance of Multi-Head Attention on MAMEM-SSVEP-II.
> |Heads |Accuracy|
> |-|-|
> |1 (Single-Head)|66.3±2.1|
> |2|67.2±2.3|
> |4|**67.6±2.5**|
>
> Dear reviewer, we have conducted extra experiments on the **MAMEM-SSVEP-II** dataset. Specifically, we compared configurations with 1 (single-head), 2, and 4 heads, respectively. As shown in Tab.B, increasing the number of heads consistently improves performance, demonstrating that multi-head configurations enable the model to capture richer interaction features. This outcome aligns with results from Euclidean multi-head attention mechanisms. We appreciate the reviewer’s insightful comments and will explore multi-head attention further in future work.

---

### Official Review · Reviewer_Q2Fj · 2024-11-03

**Soundness:** 3
**Presentation:** 4
**Contribution:** 3
**Rating:** 8
**Confidence:** 5

**Summary:**

The main contributions of this paper can be summarized as follows:

Proposal of a General Gyro Attention Framework (GyroAtt): This framework extends the classic self-attention mechanism to gyro spaces, making it applicable to various non-Euclidean matrix manifolds, such as Symmetric Positive Definite (SPD), Symmetric Positive Semi-Definite (SPSD), and Grassmann manifolds. GyroAtt unifies the construction of attention mechanisms on these manifolds. Within this framework, the authors introduce gyro homomorphisms and geodesic-based attention mechanisms to enable feature transformations and similarity calculations. The use of geodesic distances for attention scoring and weighted Fréchet mean aggregation ensures that the attention mechanism preserves the manifold's geometric structure.

Validation of the Framework’s Effectiveness Across Multiple Matrix Manifolds: The authors tested the GyroAtt framework on SPD, SPSD, and Grassmann manifolds and conducted experiments on four EEG datasets, demonstrating its excellent performance and adaptability.

**Strengths:**

The paper is clearly structured, and the mathematical framework and definitions used are rigorous and precise. In terms of methodological design, the GyroAtt framework has strong mathematical generality, overcoming the limitations of specific geometric spaces and being applicable to a variety of non-Euclidean geometries. Additionally, the implementation of GyroAtt makes full use of abstract and concise gyro homomorphisms and geodesic-based attention computation, and employs weighted Fréchet mean for aggregation, achieving a balance of simplicity and mathematical rigor.

This paper represents a continuation and innovation in modeling with gyrovector space on the SPD manifold. This approach is the first to construct an attention networks on the SPD manifold using the gyrovector space approach.

**Weaknesses:**

Primary Concerns: In the experimental section, the proposed method outperforms almost all existing methods. However, do these results reflect the best performance of the compared models? What parameters and training methods did the authors choose for these models? This question is significant because some alternative tuning may improve the performance of certain baseline models, especially in the case of several deep learning models.

Additionally, it is suggested that the authors provide more focused experimental comparisons with MAtt and GDLNet, rather than general comparisons with other models. These two models also involve optimizations of the attention mechanism on SPD manifolds and are thus the most directly comparable to the proposed method. The current design of the experiments may dilute the focus on these two models.

Possibly the most critical point: The attention structure proposed in this paper is based on scalar multiplication and vector addition in gyrovector space, rather than traditional Euclidean operations. While this network architecture differs fundamentally from the design concepts of existing methods, it also makes it challenging to directly interpret its superior performance through comparisons. Other neural networks on the SPD manifold, while not using the gyrovector space approach, have still effectively designed network structures suited for SPD matrix-valued inputs, processing non-Euclidean data by operating in the tangent space. Although the mapping between the tangent space and the manifold may introduce some error, the overall framework, being based on deep neural networks, has strong robustness that typically adjusts for such discrepancies. Based on your experimental results, there does appear to be an overwhelming performance advantage for the gyrovector space approach. This leaves me feeling puzzled. Hence, could the authors further clarify from a theoretical or practical perspective why the type of your model based on gyrovector space offers performance advantages?

The reason I refer to your results as an 'overwhelming‘ performance advantage is that even a few percentage points of average improvement on the EEG dataset is very substantial. Generally, there are always some subjects who perform poorly, which lowers the overall average. If such an 'overwhelming‘ performance advantage truly exists, then whether we should continue along the current research path focused on SPDNet for model development (such as the Graph-CSPNet and TSMNet mentioned in the paper) is a question worth further discussion. What are your thoughts on this? In terms of computational complexity and scalability, does your method also have advantages over the SPDNet pathway, or are there some potential drawbacks?

**Questions:**

1. Conduct further ablation studies or direct comparisons focused on GyroAtt, MAtt, and GDLNet, examining their performance and computational efficiency. Highlight key differences in both effectiveness and processing requirements to clarify where each model excels.

2. Provide a detailed analysis comparing the computational complexity and scalability of GyroAtt with SPDNet-based methods (maybe also be called the tangent space approach). Include a theoretical analysis or an empirical study directly comparing gyrovector operations with tangent space operations to illustrate where and why performance gains occur. Perform an ablation study that substitutes gyrovector operations with equivalent Euclidean or tangent space operations to isolate the unique impact of the gyrovector method.

3. Explain the connection between Equation (7) and the classic attention function. Are there any other possible approaches?

##Open Questions##:

What insights do you have on future research directions, including possible integrations or enhancements of SPDNet-based methods with gyrovector space concepts? Are there suggestions for future research that could merge the strengths of both approaches or further improve GyroAtt?

---

> ### Author Response · Authors · 2024-11-24
> **Response to Reviewer Q2Fj (1/3)**
>
> $\textcolor{red}{\rm{Rebuttal}}$
>
> We thank Reviewer $\textcolor{orange}{Q2Fj (R3)}$  for the constructive suggestions and insightful comments! In the following, we respond to the concerns in detail. 😄
>
> **1. These results reflect the best performance of the compared models.**
>
> - **For MAMEM-SSVEP-II and BCI-ERN.** We reported the best results from the original GDLNet and MAtt papers, applying the same preprocessing methods as described in MAtt.
>
> - **For BNCI2014001 and BNCI2015001.** We followed the preprocessing methods outlined in TSMNet. During optimization, we used the manifold optimization tools provided in the original papers for MAtt and GDLNet. We utilized the open-source code for these models and conducted extensive experiments to tune hyperparameters.
>
> **2. Clarification on the use of gyro operations in GyroAtt.**
>
> The proposed GyroAtt framework does not incorporate **gyro scalar multiplication** ($\otimes$) as part of its attention operations. Instead, it relies on two primary operations:
>
> - **Gyro Homomorphism:** This operation generalizes linear mappings from Euclidean space to gyrovector spaces. In traditional Euclidean attention mechanisms, linear transformations are applied to input data using matrix multiplication (e.g., $W x$). Gyro homomorphisms extend this concept to respect the manifold’s geometric structure, enabling linear transformations that are consistent with gyrovector space properties.
> - **Gyro Addition:** This operation generalizes the addition of a bias term in Euclidean space. In Euclidean settings, a bias $b$ is added to the transformed data (e.g., $W x + b$). Gyro addition allows the inclusion of a bias within the gyrovector space framework, ensuring consistency with the underlying manifold structure.
>
> By extending the fundamental operations of linear mapping and bias addition to gyrovector spaces through gyro homomorphism and gyro addition, GyroAtt naturally generalizes the Euclidean attention mechanism.
>
> **3.More focused experimental comparisons with MAtt and GDLNet.**
>
> Tab. 4 and 5 now provide a clearer emphasis on the performance of our GyroAtt models in direct comparison with MAtt and GDLNet. For more comparisons with MAtt and GDLNet, please refer to the common response **CQ#4**.

---

> ### Author Response · Authors · 2024-11-24
> **Response to Reviewer Q2Fj(2/3)**
>
> **4. Eq. (7) computes attention weights on manifolds.**
>
> In the classic attention function, the similarity between queries and keys is computed via dot products, capturing correlations in Euclidean space. In the GyroAtt framework, we replace the Euclidean similarity measures with the geodesic distance on the manifold. Since increased similarity correlates inversely with geodesic distance, we propose the function $(1 + \log(1 + d(Q_i, K_j)))^{-1}$ to transform geodesic distance into a valid similarity measure.
>
> One alternative is to use inner product-based similarity measures,  analogous to the dot product in Euclidean space. Following prior works [b,c], inner products can be defined on three manifolds.
>
> For $P, Q \in \mathcal{S} _{d} ^{++}, $ the SPD inner product is given by:
> $ \langle P, Q \rangle ^g = \langle Log ^g _{I _d}(P), Log ^g _{I _d}(Q) \rangle _{I _d} ^g, $
>
>
> For $ U, V \in \widetilde{\mathcal{G}} (q,d) $, the inner product is given by: $ \langle U, V \rangle ^{gr} = \langle \widetilde{Log} ^{gr}_{\widetilde{I} _{d,q}}(U), \widetilde{Log} ^{gr} _{\widetilde{I} _{d,q}}(V) \rangle _{\widetilde{I} _{d,q}}, $
>
> For $\left(U _P, S _P\right), \left(U _Q, S _Q\right) \in \widetilde{\mathcal{G}}(q,d) \times \mathcal{S} _{q} ^{++}$, the inner product is defined as: $ \langle (U _P,S _P), (U _Q,S _Q) \rangle ^{psd,g} = \lambda \langle U _P U _P ^{\top}, U _Q U _Q ^{\top} \rangle ^{gr} _{\widetilde{I} _{d,q}} + \langle S_P, S_Q \rangle ^g _{I_q}. $
>
>
> To explore this, we conducted ablation studies by substituting the distance-based similarity in Eq. (7) with the inner product. We performed experiments on Tab. A. The results show that geodesic distance generally outperforms inner product-based similarity. This is because the inner product operates in the tangent space, providing only a linear approximation around a reference point (e.g., the identity matrix $I_d$). This approximation often fails to capture relationships between points that are distant from the reference point, limiting its ability to model global relationships on the manifold. In contrast, geodesic distance measures the shortest path between two points along the manifold's curved surface, accounting for the intrinsic geometry. This allows geodesic distance to more accurately reflect the true similarities between data points on the manifold.
>
> Table A: Ablation of Similarity Measures.
>
> |Methods|Similarity|BNCI2014001 inter-session|BNCI2015001 inter-session|MAMEM-SSVEP-II|
> |-|-|-|-|-|
> |GyroAtt-SPD|Inner Product|74.7±6.8|85.6±5.4|63.9±3.2|
> |GyroAtt-SPD|Geodesic Distance|75.4 ± 7.4|86.2 ± 4.5|66.3 ± 2.2|
> |GyroAtt-Gr|Inner Product|72.4 ± 7.3|83.4 ± 5.9|65.7 ± 3.1|
> |GyroAtt-Gr|Geodesic Distance|72.5 ± 7.3|85.0 ± 7.7|67.1 ± 1.6|
> |GyroAtt-SPSD|Inner Product|71.6 ± 6.3|83.3 ± 5.4|65.0 ± 2.6|
> |GyroAtt-SPSD|Geodesic Distance|72.9 ± 6.2|85.3 ± 5.3|68.7 ± 1.5|

---

> ### Author Response · Authors · 2024-11-24
> **Response to Reviewer Q2Fj(3/3)**
>
> **5. Regarding future research directions**
>
> We appreciate the reviewer's thoughtful question about the future of research in this area. The research path represented by SPDNet has developed numerous operations and achieved notable success, with core components primarily based on BiMap, ReEig, and LogMap layers. BiMap and ReEig are largely numerical operations that do not fully exploit Riemannian geometry. LogMap, on the other hand, is a Riemannian logarithm operation based on the tangent space at the identity matrix under the Affine-Invariant Riemannian Metric (AIRM) or Log-Euclidean Metric (LEM). However, linearization methods relying on a single fixed tangent space or coordinate system may not effectively capture the global geometric properties of the manifold.
>
> Recent advances have shown promising directions. [e,f] have proposed constructing gyrovector spaces on hyperbolic manifolds using corresponding Riemannian operators, leading to the development of Multinomial Logistic Regression (MLR), fully-connected (FC), and convolutional layers suitable for hyperbolic manifolds. Fully hyperbolic convolutional neural networks have also been introduced [g]. Additionally, [b,c] have extended these concepts to SPD manifolds, constructing MLR in the gyrovector space of SPD manifolds. [a,d] have progressed further by overcoming the limitations of gyro structures and directly implementing MLR on manifolds using Riemannian logarithms and related operators.
>
> These developments provide valuable insights. In future research on SPD deep learning networks, we may need to place greater emphasis on the intrinsic geometric properties of SPD manifolds, rather than solely maintaining their SPD attributes. Integrating the strengths of SPDNet-based methods with gyrovector space concepts could lead to more powerful and geometrically faithful models.
>
>
> **References**
>
> > [a] RMLR: Extending multinomial logistic regression into general geometries. In NeurIPS 2024.
> >
> > [b] Building Neural Networks on Matrix Manifolds: A Gyrovector Space Approach. In ICML 2023.
> >
> > [c] Matrix manifold neural networks++. In ICLR 2024.
> >
> > [d] Riemannian multinomial logistics regression for SPD Neural Networks.
> >
> > [e] Hyperbolic neural networks. In NeurIPS 2018.
> >
> > [f] Hyperbolic neural networks++. In ICLR 2021.
> >
> > [g] Fully hyperbolic convolutional neural networks for computer vision. In ICLR 2024.

---

> ### Comment · Reviewer_Q2Fj · 2024-11-25
>
> Thank you for your detailed response! I’ve already given you added points.
>
> You’ve made an excellent point: tangent space is inherently a local geometric concept. Therefore, when using tangent space methods, we must always consider the issue of choosing a base point. Your research approach effectively addresses this limitation to some extent, which is highly commendable.
>
> At the same time, I am curious if it would be possible to further explore the potential improvements that the global geometric properties you mentioned might bring to solving EEG classification problems. From my perspective, your work stands out for its mathematical completeness. However, finding a stronger connection between the theoretical aspects and tangible improvements in task performance could be a worthwhile direction to explore. This linkage might enhance the engineering value and application potential of your research even further.

---

> ### Author Response · Authors · 2024-11-25
> **Thanks for your instant replay**
>
> Thanks for the instant replay and encouraging feedback!
>
> The Gyro space naturally extends vector spaces to manifolds, offering convenient mathematical tools to construct Riemannian networks. We have conducted ablation studies to validate each basic block of GyroAtt, including the activation (Sec. 6/Tab.6), and transformations via homomorphism (Sec. B.4). All these operations respect the latent matrix geometries and their effectiveness has been validated by ablations.
>
> In the future, we plan to scale gyro networks for larger datasets and deeper models, such as the TUAB [a] and TUEV [a] datasets, as well as their integration with the large model (LaBraM)  [b].
>
>
>
> If you have any additional comments or suggestions regarding our empirical analysis, we would be delighted to hear them! 😄
>
> **References**
>
> > [a] The Temple University Hospital EEG data corpus. In Frontiers in Neuroscience 2016.
> >
> >[b] Large brain model for learning generic representations with tremendous EEG data in BCI. In ICLR 24

---

### Official Review · Reviewer_zEZt · 2024-11-04

**Soundness:** 3
**Presentation:** 3
**Contribution:** 2
**Rating:** 6
**Confidence:** 2

**Summary:**

The paper presents GyroAtt, a novel attention framework on matrix manifolds that unifies various manifold-based attention mechanisms. The framework generalizes attention operations to gyrovector spaces, incorporating specific implementations for SPD, SPSD, and Grassmannian manifolds. Experiments show competitive performance on EEG datasets compared to traditional methods, demonstrating both the versatility and effectiveness of the GyroAtt framework.

**Strengths:**

1. The proposed GyroAtt framework provides a unified approach to manifold-based attention mechanisms, which helps in applying attention across various matrix manifolds.
2. The framework’s flexibility is validated by empirical performance across multiple gyro structures, offering a robust approach for EEG data applications.
3. The GyroAtt framework shows strong results on EEG datasets, outperforming or being competitive with existing methods, which is promising for non-Euclidean data.

**Weaknesses:**

1. Although the framework generalizes attention, it primarily applies existing forms, such as SPD, Grassmannian, and SPSD manifolds, limiting exploration into novel or unknown structures.
2. As noted in Tables 4 and 5, the performance improvements are not consistently significant across datasets, and some results fall within the margin of error.

**Questions:**

1. Given that the framework mainly specifies known forms (SPD, Grassmannian, SPSD), can GyroAtt be extended to unknown or less conventional structures? If not, could the authors clarify the contribution of GyroAtt as primarily a unifying formulation rather than a framework for discovering new structures ?

2. There are symbols introduced without clear definitions in the text, like the commutator [A, B] used in line 92 and its meaning only clarified much later in line 146. A glossary or early explanation could improve clarity .

3. Eq. (9) introduces an additional layer to increase model expressivity. Could the authors explain its functional relationship to the Gyro-attention block? Is this added primarily to improve empirical performance, or does it hold a theoretical grounding within the framework?

4. Symbols like \downarrow in Theorems 5.1 to 5.4 and parentheses in Equation (16) may be specialized to certain subfields but are not well-defined within the text. Briefly introducing these notations in the initial sections would enhance readability.

5. In Table 4, GyroAtt-SPSD achieves the best performance, while GyroAtt-SPD is better in Table 5. Could the authors provide an intuitive explanation for why different variants excel on different datasets?

6.  The standard deviations in Tables 4 and 5 suggest that performance differences may not always be statistically significant.

7. Understanding the limitations and computational complexity of GyroAtt would be useful for researchers considering this method for resource-intensive applications. Could the authors include an analysis of these aspects, particularly comparing them with baseline methods?

---

> ### Author Response · Authors · 2024-11-24
> **Response to Reviewer zEZt (1/3)**
>
> $\textcolor{red}{\rm{Rebuttal}}$
>
> We thank Reviewer $\textcolor{red}{zEZt (R2)}$ for the careful review and the suggestive comments. Below, we address the comments in detail. 😄
> ***
>
> **1. GyroAtt can generalize to novel or unknown Gyrovector Spaces.**
>
> The GyroAtt framework is inherently adaptable and versatile, as described in Sec. 4 of our paper. In given gyrovector spaces, the primary requirement for GyroAtt is the availability of a weighted Fréchet mean (WFM), a widely applicable operation on various manifolds. By incorporating the appropriate gyro homomorphisms and WFM, GyroAtt can be extended to any gyrovector space, including unexplored geometries such as hyperbolic manifolds [a]. Notably, to the best of our knowledge, we are the first to propose an attention mechanism specifically for the SPSD manifold. While GyroAtt is theoretically generalizable to other geometric structures, this study focuses on matrix manifolds, such as SPD, SPSD, and Grassmannian, due to their proven effectiveness in EEG applications.
>
> **2. The symbol gyr$[A, B]$ denotes the gyroautomorphism or gyration generated by elements $A$ and $B$.**
>
> We define gyr$[A, B]$ at line 93 in the revised manuscript and a summary of the used symbols in this manuscript has been provided in App. A. Specifically, gyr$[A, B]$ denotes the gyroautomorphism or gyration generated by elements $A$ and $B$ in a gyrogroup.

---

> ### Author Response · Authors · 2024-11-24
> **Response to Reviewer zEZt (2/3)**
>
> **3. Theoretical and Empirical analysis for the Bias and Activation Layer in GyroAtt.**
>
> (1) **Theoretical Perspective**. Eq. (9) introduces the Gyro addition $\oplus$ and a matrix power-based nonlinear activation function $\sigma$, designed to manipulate the metric spaces and maintain numerical stability within the GyroAtt framework. Unlike the ReEig function commonly used in existing SPD networks—which primarily serves as a numerical stabilization method by normalizing small eigenvalues to a slightly larger positive constant—our power-based activation function directly integrates with the Riemannian geometry of the manifold. This function not only preserves the SPD structure but also inherently regularizes the Riemannian metric, aligning with the latent geometric properties of the manifold.
>
> Moreover, as illustrated in [d] (Fig. 1), the power operation provides a unique capability to transition between different Riemannian metric geometries. For example, when the parameter $\theta$ approaches zero, the Affine-Invariant Metric (AIM) will transition into the Log-Euclidean Metric (LEM). This dynamic adjustment enables the activation function to adapt to different geometric contexts while maintaining stability.
>
> From a numerical perspective, the power-based activation could effectively control the condition number of the matrix by modulating its eigenvalues, thus mitigating the effects of ill-conditioning. Besides, the Gyro addition-based bias operation can dynamically adjust the distribution of eigenvalues, not only enabling the model to adapt to different feature distributions but also further enhancing the efficacy of the power function. This dual benefit—improved numerical stability and alignment with manifold geometry—distinguishes our approach from ReEig, which lacks the flexibility to encode such geometric transformations.
>
> (2) **Empirical analysis.** The ablation results are presented in Tab. A and B show the impact of Gyro addition and power-based activation in Eq. (9) on the accuracy of GyroAtt-SPD and GyroAtt-SPSD across three used datasets. Wherein, 'w/o' and 'w/'' indicate that the respective component is excluded and included in the model, respectively. The results indicate that incorporating both the bias operation and the nonlinear activation function leads to the improved classification ability for both GyroAtt-SPD and GyroAtt-SPSD models. We argue that the fundamental reasons lie in two aspects: 1) the power-based activation enhances the nonlinear representational capacity of the designed model; 2) regularizing the metric space can make the learned feature manifold more consistent with the space assumed by the classifier. In other words, the regularization to the metric space can reduce the geometric distortion when embedding manifold-valued data into Euclidean space, thus improving the learning ability of the classifier.
>
> Table A: Accuracy (\%) comparison of GyroAtt-SPD under different combinations of $\oplus$ and $\sigma$.
> |Bias$(\oplus)$|Activation$(\sigma)$|BNCI2014001 Inter-session| BNCI2015001 Inter-session |MAMEM-SSVEP-II|
> |-|-|-|-|-|
> |w/o|w/|74.7 ± 7.0|85.6 ± 5.1|65.2 ± 2.3|
> |w/|w/o|74.8 ± 6.7|85.4 ± 5.0|64.3 ± 2.5|
> |w/o|w/o|74.4 ± 7.5|85.3 ± 5.2|64.5 ± 2.6|
> |w/|w/|75.4 ± 7.4|86.2 ± 4.5| 66.3 ± 2.2|
>
> Table B: Accuracy (\%) comparison of GyroAtt-SPSD under different combinations of $\oplus$ and $\sigma$.
> |Bias$(\oplus)$|Activation$(\sigma)$|BNCI2014001 Inter-session|BNCI2015001 Inter-session|MAMEM-SSVEP-II|
> |-|-|-|-|-|
> |w/o|w/|72.5 ± 6.7|84.9 ± 6.2|67.3 ± 2.5|
> |w/|w/o|72.2 ± 7.2|83.9 ± 5.1|66.5 ± 2.4|
> |w/o|w/o|72.0 ± 6.8|84.1 ± 6.1|66.2 ± 2.3|
> |w/|w/|72.9 ± 6.2|85.3 ± 5.3|68.7 ± 1.5|
>
> **4. The symbol $\downarrow$ links to the corresponding theoretical proof in the appendix.**
>
> The symbol $\downarrow$ is a clickable reference linking to the corresponding theoretical proof in the appendix. To enhance clarity, we have added an explanation of this symbol in the main text.

---

> ### Author Response · Authors · 2024-11-24
> **Response to Reviewer zEZt (3/3)**
>
> **5. The differing performance between GyroAtt-SPD and GyroAtt-SPSD likely stems from numerical instability in SPD-based operations.**
>
> The varying performance between GyroAtt-SPD and GyroAtt-SPSD across datasets is likely due to differences in dataset characteristics and manifold structures. In EEG covariance modeling, rank deficiency is a common issue that impacts the positive definiteness required by SPD manifolds. Both GyroAtt-SPD and GyroAtt-SPSD are affected by the numerical instability inherent in SPD-based operations (e.g., SVD, eigen decomposition). One contributing factor is that network inputs are often ill-conditioned. To mitigate this, we use a custom backpropagation mechanism that approximates the gradients of SPD operations, although this may influence performance.
>
> In contrast, our SPSD representation decomposes the manifold into a subspace and an SPD matrix. While the SPD matrix may not always be strictly positive definite in practice, this decomposition reduces the reliance on custom backpropagation in certain steps, potentially improving performance. This likely explains why the GyroSpsd++ variant outperforms GyroSpd++ in some scenarios.
>
> Notably, these results also highlight the generality of our GyroAtt framework, which enables the selection of manifold-specific attention mechanisms based on dataset characteristics and task requirements. This flexibility is a key strength of GyroAtt, allowing it to adapt to diverse scenarios that require different geometries and attention strategies.
>
> **6. The large std is attributed to the characteristics of the EEG datasets.**
>
> On the BCI-ERN and MAMEM-SSVEP-II datasets, GyroAtt-SPSD outperforms the baseline GDLNet by 3.2% and 0.9%, respectively, while exhibiting lower standard derivations, underscoring its superior performance and robustness.
>
> For the BNCI2014001 and BNCI2015001 datasets, GyroAtt-SPD achieves notable improvement in terms of accuracy and std over other Riemannian-based methods. Although the standard deviations are relatively higher, this can be attributed to the intrinsic variability of EEG data, which is influenced by factors such as a low signal-to-noise ratio (SNR), significant domain shifts, and low specificity in inter-session and inter-subject scenarios. These challenges highlight the practical difficulty of achieving consistent performance across such datasets.
>
> **7. The limitation of GyroAtt is its specificity to Gyrovector spaces, restricting its applicability to manifolds lacking similar structures.**
> A limitation of GyroAtt is its specificity to gyrovector spaces, which restricts its applicability to manifolds lacking gyro structures. Its design is inherently tied to the algebraic properties of gyrovector spaces. Another limitation lies in its computational complexity. For specific metrics and manifolds, such as GyroAtt-AIM, the computational cost of GyroAtt can be higher compared to other manifold-based attention methods, as detailed in the common response **CQ#3**.
>
> **References**
>
> >[a] Hyperbolic Neural Networks++. In ICLR 2021.
> >
> >[b] Matrix Backpropagation for Deep Networks with Structured Layers. In ICCV,2015.
> >
> >[c] MAtt: A manifold attention network for EEG decoding.
> >
> >[d] SPD domain-specific batch normalization to crack interpretable unsupervised domain adaptation in EEG. In NeurIPS, 2022.
> >
> >[e] Matrix Manifold Neural Networks++. In ICLR, 2024.

---

> > ### Comment · Reviewer_zEZt · 2024-11-26
> >
> > I greatly appreciate the authors’ detailed response and have no further questions at this time.

---

> ### Author Response · Authors · 2024-11-26
> **Thank you for your prompt reply**
>
> Thank you for your prompt and positive reply! If you have any further suggestions or insights, we would be very glad to hear them! 😄

---

> ### Author Response · Authors · 2024-11-28
> **Ablations on matrix power and condition numbers**
>
> Dear reviewer, we have conducted additional ablation on the condition numbers that we briefly mentioned before.
>
> **The power-based activation function can effectively reduce matrix condition numbers.**
>
> We randomly sample 800 SPD matrices from the MAMEM-SSVEP-II dataset and 720 from the BCI-ERN dataset. We focus on the GyroAtt-SPD backbone and measure the condition numbers of input and output of the activation layers, where activations could be our matrix power or the ReEig.
>
> Table C: Condition number （$\kappa$） of input and output SPD matrices of the power activation layer on the MAMEM-SSVEP-II dataset.
> |power|$\kappa<100$|$100<\kappa<1000$|$1000<\kappa<5000$|$5000>\kappa$|
> |-|-|-|-|-|
> |Input|0|508|292|0|
> |Output|730|70|0|0|
>
> Table D: Condition number （$\kappa$） of input and output SPD matrices of the ReEig layer layer on the MAMEM-SSVEP-II dataset.
> |ReEig|$\kappa<100$|$100<\kappa<1000$|$1000<\kappa<5000$|$\kappa>5000$|
> |-|-|-|-|-|
> |Input|0|508|79|721|
> |Output|0|8|666|126|
>
> Table E: Condition number （$\kappa$） of input and output SPD matrices of the ReEig layer layer on BCI-ERN dataset.
> |power|$\kappa<100$|$100<\kappa<1000$|$1000<\kappa<5000$|$\kappa>5000$|
> |-|-|-|-|-|
> |Input|3|656|61|0|
> |Output|720|0|0|0|
>
> Table F: Condition number （$\kappa$） of input and output SPD matrices of the ReEig layer layer on the BCI-ERN dataset.
> |ReEig|$\kappa<100$|$100<\kappa<1000$|$1000<\kappa<5000$|$\kappa>5000$|
> |-|-|-|-|-|
> |Input|0|461|259|0|
> |Output|174|546|0|0|
>
>
>
> As shown in the tables, both the ReEig layer and the power-based activation function reduce condition numbers, but the power-based activation performs more effectively. On **MAMEM-SSVEP-II**, the power-based activation reduces all condition numbers to $\kappa < 1000$, with 730 matrices achieving $\kappa < 100$. In contrast, ReEig leaves many matrices with higher condition numbers. Similarly, on **BCI-ERN**, the power-based activation results in 720 matrices with $\kappa < 100$, while ReEig retains matrices with $\kappa$ values in the range $100 < \kappa < 1000$. The power-based activation provides a more robust approach to reducing condition numbers, ensuring that no extreme outliers with high $\kappa$ values remain. This contrasts with the ReEig layer, which often leaves matrices with high condition numbers, highlighting the superior ability of the power-based activation to control matrix conditioning.
>
> **In-depth Analysis.** The power-based activation function effectively controls the condition number when the exponent $|p|$ in the power-based activation is less than 1. The condition number of a matrix is typically defined as the ratio of its largest to smallest eigenvalue, $\frac{\lambda_{max}}{\lambda_{min}}$. The power transformation reduces the influence of larger eigenvalues while amplifying smaller ones. After applying the power-based activation function, this ratio becomes $\left( \frac{\lambda_{max}}{\lambda_{min}} \right)^{p}$, where $p < 1$. Furthermore, this function amplifies the influence of smaller eigenvalues, thereby facilitating the learning of latent information within them.

---

### Official Review · Reviewer_bQUv · 2024-11-05

**Soundness:** 4
**Presentation:** 3
**Contribution:** 2
**Rating:** 5
**Confidence:** 4

**Summary:**

This article introduces an abstract framework to build attention layers
in various Riemannian manifolds. It generalizes a few previous works
suitable only for particular geometries, namely SPD, SPSD and
Grassmannian manifolds. The proposed layer are validated on somme EEG
classification tasks

**Strengths:**

A lot of technical work is presented in the article, it shows a mastery
of the mathematical tools and I am always supportive of this kind of
generalization work. The experiments are interesting and shows the
interest of Riemannian networks in particular applications.

**Weaknesses:**

The writing is barely understandable. I have the feeling that the
authors try to find by all means the way to put as much as possible
content in the 10 pages. A lot of things are relegated to appendices, so
a reader will only have references in the main text: it's not acceptable
to simply refer to for example "Alg 3" in a table, without even a word
about it in the text (I know most of them are Karcher flows, but still,
just write Karcher flow in this case). I disagree with the expression
"More details are given in annex": it's "more", it's simply "all"...

The authors also play a dangerous game with line spacing (for example
lines 94-100, but in other places). Even if it does not reach (probably)
the level for a desk reject, it's unreadable and disrespectful for the
reader.

There are too much abbreviations.

**Questions:**

It lacks a convincing motivation for the generalization. Does it give a
better understanding of the existing layers ? Does it permit to build
new layer for other manifolds ?

It's not clear for me if the generalization applied to a particular
geometry is strictly equivalent to the specific layer existing in prior
work or if it's just similar.

What is the precise architecture used in the experiments ? There is a
short description but more details must be given since a lot of tricky
and critical choices are made at this level. It is not sufficient to put
5 lines in a paragraph called "Implementation details". In addition to a
precise description, a drawing of the architecture is often a good idea.

I am not sure if the long discussion on the influence of the coefficient
in the power activation layer is really necessary. In general it's
certainly interesting but I have the feeling it's not the priority here
due to the lack of space.

What is the metric used for SPD experiments in Table 4 ?

---

> ### Author Response · Authors · 2024-11-24
> **Response to Reviewer bQUv (1/2)**
>
> $\textcolor{red}{\rm{Rebuttal}}$
>
> We thank Reviewer $\textcolor{blue}{bQUv (R1)}$ for the careful review and the suggestive comments. Below, we address the comments in detail. 😊
> ***
>
> **1. Writing: Reorganized Structure, Improved Line Spacing, and Reduced Abbreviations.**
>
> We thank the reviewer for their comments on the writing. We have made significant revisions to the manuscript, which are summarized below.
>
> - **Reorganized structure and implementation details.** We appreciate the reviewer’s suggestion regarding Table 7 and its related ablation study. As we recognize this is secondary to the main content, we have moved it to App. B. To enhance the clarity of the main text, we have added more implementation details and included a model diagram (Figure 1) to provide a clearer visualization of the GyroAtt architecture. Additionally, The Karcher flow algorithm is now included in Tab. 3. We have concisely summarized other appendix content and provided appropriate references in the main paper.
> - **Addressed Line Spacing Issue.** We thank the reviewer for pointing out the line spacing issue between lines 94-100. We understand that it may have impacted the reading experience, while this formatting was automatically generated by the ICLR 2025 template and not manually adjusted. To address this, we have restructured the content in this section and clarified the text for a smoother flow. We hope these changes meet the reviewer’s expectations.
> - **Reduced abbreviations and added a list of abbreviations in App. A.** We thank the reviewer for highlighting the issue of excessive abbreviations. In response, we have further reduced unnecessary abbreviations throughout the manuscript to improve clarity. A list of abbreviations is included in App. A and recapped below.  Apart from commonly used terminologies in matrix manifolds, the revised manuscript retains only one additional abbreviation: GyroAtt.
>
> Table A: Summary of Abbreviations.
> |Abbreviations|Explanation|
> |-|-|
> |SPD|Symmetric Positive Definite|
> |SPSD|Symmetric Positive Semi-Definite|
> |GyroAtt|Gyro Attention (our network name)|
> |EEG|Electroencephalography|
> |LEM|Log-Euclidean Metric|
> |LCM|Log-Cholesky Metric|
> |AIM|Affine-Invariant Metric|
> |WFM|Weighted Fréchet Mean|
> |ONB|Orthonormal Basis|
>
> **2. Motivations.**
>
> - **From Euclidean vector spaces to Gyrovector Spaces.**
> Gyrovector spaces are a natural extension of vector spaces, which offers convenient mathematical tools to extend Euclidean networks into manifolds. They have shown success in different manifolds, such as [c,d,e].
>
> - **From specific to principled manifold attention.** Previous work, such as MAtt [a] and GDLNet [b], focuses on establishing attention on specific manifolds, although they have shown success in their interested manifolds. For instance, the transformation (BiMap) layer in MAtt is exclusive to the SPD manifold. Therefore is no clue how to design BiMap-like layers in other manifolds. A similar issue also happens to GDLNet. In contrast, our GyroAtt resorts to the gyro space, which can design basic layers, including transformation, attention calculation, and aggregation, in a principled manner. Empirically, we manifest our GyroAtt on three matrix manifolds in a principled manner, including SPD, Grassmannian, and SPSD manifolds covering seven different gyro spaces.
>
> - **From linear layer to gyro homomorphism.** Gyro homomorphism naturally generalizes linear layers from Euclidean to gyrovector spaces. When the gyrovector space reduces to Euclidean space, the gyrogroup operations $\oplus$ and $\otimes$ simplify to standard vector addition and scalar multiplication. Under these simplifications, the gyro homomorphism naturally degenerates into a standard linear mapping, maintaining consistency with traditional Euclidean operations.
>
>
> Building on these points, we propose a general framework for attention computation in gyrovector spaces, called **GyroAtt**.
>
> **3. Difference with previous manifold attention**
>
> For differences between GyroAtt, MAtt, and GDLNet, please refer to the **CQ#2** in Common Response.
>
> **4. GyroAtt permits the construction of new layers for other manifolds.**
>
> As described in Sec. 5, GyroAtt implements attention mechanisms on the SPD, SPSD, and Grassmannian manifolds. Notably, to the best of our knowledge, we are the first to propose an attention mechanism on the SPSD manifold. As shown in Sec. 4, Our GyroAtt framework comprises three basis layers: gyro homomorphism, WFM, and bias. the bias and gyro homomorphism are derived naturally from the properties of gyrovector spaces. By incorporating the appropriate WFM, GyroAtt can be extended to strict or generalized gyrovector spaces, including hyperbolic manifolds [e].

---

> > ### Comment · Reviewer_bQUv · 2024-11-25
> >
> > Thanks for the updates, the article is much more readable by now, I will
> > update by marks accordingly.
> >
> > In addition to the presentation improvements, I appreciate:
> > - the description of the architecture
> > - the comment on the consistency with Euclidean geometry

---

> ### Author Response · Authors · 2024-11-24
> **Response to Reviewer bQUv (2/2)**
>
> **5. We have added a detailed architecture figure and expanded the implementation details in the revised manuscript.**
>
> For a detailed comparison and explanation, please refer to **CQ#1** in the common response.
>
> **6. We have moved Tab. 7 and the related discussions to the appendix, as suggested, to better manage space constraints.**
>
> We appreciate your feedback regarding the discussion on the influence of the coefficient in the power activation layer. We have revised the ablation study accordingly. We have retained Tab. 6 in the main text as it highlights the impact of different geometries—AIM, LEM, and LCM—on the performance of GyroAtt-SPD and GyroAtt-SPSD. This aspect is central to our analysis, as it directly ties to the primary contributions of our work, while the analysis of activation functions serves as a secondary aspect.
>
> **7. The SPD experiments in Tab. 4 are respectively equipped with AIM and LEM on the MAMEM-SSVEP-II and BCI-ERN datasets.**
>
> In Tab. 4, the metrics for GyroAtt-SPD are configured as follows: **AIM** on the MAMEM-SSVEP-II dataset and **LEM** on the BCI-ERN dataset. In Tab. 5, which presents results on the BNCI2014001 and BNCI2015001 datasets, **AIM** is used for both. We have updated the manuscript to include these details, clarifying the metrics used in our SPD experiments.
>
> To provide a comprehensive view, we present the performance of GyroAtt-SPD under different metrics across all datasets in the table below:
>
>
> Table B: The results of GyroAtt-SPD under different metrics across all the involved datasets
> |Metric|MAMEM-SSVEP-II|BCI-ERN| BNCI2014001 Inter-session|BNCI2014001 Inter-subject|BNCI2015001 Inter-session|BNCI2015001 Inter-subject|
> |-|-|-|-|-|-|-|
> |AIM|**66.3±2.2**|75.1±3.4|**75.4±7.1**|**53.1±14.8**|**86.2±4.5**|**77.9±13.0**|
> |LEM|66.2±2.5|**76.1±4.2**|75.3±6.5| 52.3±14.1|85.7±5.5|76.6±13.7|
> |LCM|65.1±2.5|75.3±4.5| 74.2±7.8|52.7±13.6|86.0±6.8|77.4±13.2|
>
> The results indicate that AIM consistently outperforms LEM and LCM in most cases. This is mainly attributed to the AIM's affine invariance, a crucial property for EEG-based Brain-Computer Interface (BCI) applications. As analyzed in [f], preprocessing steps such as re-referencing, filtering, and normalization often induce affine transformations to the constructed covariance matrices. The invariance of AIM preserves the intrinsic geometric properties of these data points, enhancing the robustness of such transformations. In contrast, LEM approximates the global geometry of the SPD manifold, while LCM lacks the ability to model the manifold's global characteristics. Both of the above situations are not conducive to learning effective EEG features.
>
> **References**
>
> >[a] MAtt: A manifold attention network for EEG decoding. In NeurIPS 2022.
> >
> >[b] A Grassmannian manifold self-attention network for signal classification. In IJCAI 2024.
> >
> >[c] Building Neural Networks on Matrix Manifolds: A Gyrovector Space Approach. In ICML 2023.
> >
> >[d] Matrix Manifold Neural Networks++. In ICLR 2024.
> >
> >[e] Hyperbolic Neural Networks++. In ICLR 2021.
> >
> >[f] Multiclass brain-computer interface classification by Riemannian geometry. In IEEE TBME 2012.

---

> ### Author Response · Authors · 2024-11-25
> **Official Comment by Reviewer bQUv**
>
> Thank you for your prompt reply and kind feedback! If you have any additional comments or suggestions, we would be very happy to hear them! 😄

---

### Author Response · Authors · 2024-11-24
**Common Response (1/3)**

$\textcolor{red}{\rm{Rebuttal}}$

We thank all the reviewers for their constructive suggestions and valuable feedback. Below are our responses to the common questions (CQs). 😄

***

**CQ#1: The overall model structure.** $\textcolor{blue}{bQUv (R1)}$, $\textcolor{purple}{nEzp (R4)}$

In the revised manuscript, we have included the network architecture of the proposed GyroAtt in Figure 1 to offer a clear, intuitive understanding of our model. Furthermore, we have expanded the part of Implementation Details with a more comprehensive description.

**CQ#2: Differences between GyroAtt, MAtt, and GDLNet.** $\textcolor{blue}{bQUv (R1)}$, $\textcolor{purple}{nEzp (R4)}$

To highlight the distinctions among GyroAtt, MAtt, and GDLNet, we provide Tab. A, which summarizes their key differences in terms of geometry, transformation, distance metric, attention computation, and feature aggregation. Unlike prior works, GyroAtt delivers **broader generality and flexibility, adheres to gyro algebraic structures**, and ensures **consistency with Euclidean Attention Mechanisms**


Table A: Summary of several representative attention methods on different geometries.
|Methods|Geometry|Transformations|$\text{dist}(\vec{q}_i,\vec{k}_i)$|Attention $\mathcal{A}_{ij}$|Aggregation $\vec{r}_i (\vec{R}_i)$|
|-|-|-|-|-|-|
|Transformer [a] |Euclidean|$\text{Linear}(\vec{x} _i)$|$\|\|\vec{q} _{i}-\vec{k} _j\|\| _{\mathrm{F}}$|$\text{Softmax}\left(\frac{\langle \vec{q} _{i}, \vec{k} _{j} \rangle}{\sqrt{d _k}}\right)$|Arithmetic mean $\sum^{N} _{j} A _{ij} \vec{v} _{j}$|
|MAtt [b]|SPD under LEM|$WX _i W ^{T}$| $\|\left\|\text{logm}(Q _i)-\text{logm}(K _i) \right\|\| _{\mathrm{F}}$|$\text{Softmax} \left( \left( 1 + \log(1+\text{dist}(Q _{i}, K _{j})) \right) ^{-1} \right)$|LEM-based WFM $\text{expm} \left( \sum _{j=1} ^{N} \mathcal{A} _{ij} \text{logm}(V _j) \right)$|
|GDLNet [c]|Grassmannian under ONB| $WX _i$|$\left\|\left\| Q _{i}Q _{i}^{\top} - K _{j}K _j ^{\top} \right\|\right\| _{\mathrm{F}}$|$\text{Softmax} \left( \left( 1 + \log(1+\text{dist}(Q _{i}, K _{j})) \right) ^{-1} \right)$|Extrinsic WFM $\Phi ^{-1}\left( \sum _{j=1}^{N} \mathcal{A} _{ij} \Phi(V _{j}) \right)$|
| **GyroAtt (Ours)**|SPD, SPSD, Grassmannian|$\text{hom}(X _i)$|Geodesic distance|$\text{Softmax} \left( \left( 1 + \log(1+\text{dist}(Q _{i}, K _{j})) \right) ^{-1} \right)$|WFM|

- **Generality and Flexibility of GyroAtt.** GyroAtt offers a unified framework that supports multiple manifolds and metrics, including SPD, SPSD, and Grassmannian manifolds, demonstrating its superior generality and flexibility. In contrast, prior methods are typically limited to specific manifolds and metrics; for example, MAtt is restricted to the SPD manifold with the Log-Euclidean Metric (LEM), while GDLNet is tailored for the Grassmannian manifold under an orthonormal basis (ONB). Our approach enables attention mechanisms across different geometries, enhancing versatility for various applications.

- **Natural Extension of the Euclidean operations:** When the gyrovector space reduces to Euclidean space, the gyrogroup operations $\oplus$ and $\otimes$ are reduced to standard vector addition and scalar multiplication. Besides, the gyro homomorphism is reduced to the familiar linear map. This demonstrates that our definitions of homomorphism and biasing are natural extensions of the Euclidean linear map and biasing.

---

> ### Author Response · Authors · 2024-11-24
> **Common Response (2/3)**
>
> **CQ#3: Computational complexity and runtime comparison.** $\textcolor{red}{zEZt (R2)}$, $\textcolor{orange}{Q2Fj (R3)}$
>
> The computational complexity of GyroAtt, MAtt, and GDLNet is primarily driven by the operations of weighted Fréchet mean (WFM), gyro homomorphism, and similarity measurement. Among these, the type and the number of matrix functions are the main computational cost of the aforementioned manifold-valued operations.
>
> **Number of Matrix Functions in Attention Mechanisms.** It can be analyzed that the three manifold attention networks—MAtt, GDLNet, and our proposed GyroAtt—primarily involve two types of matrix functions: Singular Value Decomposition (SVD) and Cholesky decomposition. Both have an computational complexity of $\mathcal{O}(d^3)$, where $d$ represents the dimension of the matrices involved. Since GyroAtt-Gr and GyroAtt-SPSD involve SVD computations on the matrices of size $d \times q$, they have a complexity of $\mathcal{O}(d q^2)$. We summarize the number of required matrix functions for each method in Tab. B, where $C$ denotes the number of inputs to the attention module (e.g., the number of queries or keys), and $q$ is a parameter related to the subspace dimension.
>
> Table B: Complexity comparison of different manifold attention models, where "Chol" denotes the Cholesky decomposition.
> |Model|Metric|SVD (d×d) $\mathcal{O}(d^3)$|Chol (d×d) $\mathcal{O}(d^3)$|SVD (d×q) $\mathcal{O}(dq^2)$|SVD (q×q) $\mathcal{O}(q^3)$|Chol (q×q) $\mathcal{O}(q^3)$|
> |-|-|-|-|-|-|-|
> |MAtt|LEM|$3C$|0|0|0|0|
> |GDLNet|ONB|$C$|0|0|0|0|
> |GyroAtt-Gr|ONB|0|0|$C^2+C$|$2C$|0|
> |GyroAtt-SPD|AIM|$2C^2+3C$|0|0|0|0|
> |GyroAtt-SPD|LEM|$4C$|0|0|0|0|
> |GyroAtt-SPD|LCM|$C$|$3C$|0|0|0|
> |GyroAtt-SPSD|AIM|0|0|$C^2+C$|$3C^2+4C$|0|
> |GyroAtt-SPSD|LEM|0|0|$C^2+C$|$C^2+6C$|0|
> |GyroAtt-SPSD|LCM|0|0|$C^2+C$|$C^2+2C$|$3C$|
>
>
> **Experimental comparison of time efficiency.** For an intuitive comparison, we have measured the average training time (s/epoch) of our GyroAtt variants and the baseline models (MAtt and GDLNet) across the BNCI2014001 and BNCI2015001 datasets. From Tab. C, we can see that the runtime of GyroAtt-Gr is higher than that of GDLNet, primarily due to the parameter $q$ being set to 18 (nearly half of $d$) on these datasets, resulting in relatively higher computational complexity. Moreover, as inferred from Tab. B and confirmed by Tab. C, the computational efficiency of GyroAtt-SPSD under any metric is lower than that of GDLNet and MAtt. In contrast, GyroAtt-SPD with the LCM achieves a slightly faster runtime compared to MAtt. Besides, it also can be seen from Tab. C that the runtime of any GyroAtt variant under LCM is lower than that under AIM and LEM. These experimental situations are mainly attributed to the fact that Chol is typically more efficient in practice due to smaller constant factors compared to SVD. Another interesting observation in Tab. C is that the computation speed of the AIM-based GyroAtt is slower than its LEM-based counterpart, primarily because AIM requires more eigenvalue operations than LEM.
>
> Table C: Training time (s/epoch) comparison of different manifold attention networks on the BNCI2014001 and BNCI2015001 datasets.
> |Methods|Metrics|BNCI2014001 inter-session|BNCI2014001 inter-subject|BNCI2015001 inter-session|BNCI2015001 inter-subject|
> |-|-|-|-|-|-|
> |MAtt|LEM|4.86|89.12|2.74|56.78|
> |GDLNet|ONB|4.55|88.59|1.71|47.66|
> |GyroAtt-Gr|ONB|5.28|100.20|1.98|54.07|
> |GyroAtt-SPD|AIM|10.36|149.14|4.44|81.73|
> |GyroAtt-SPD|LEM|7.89|138.59|4.28|80.66|
> |GyroAtt-SPD|LCM|4.11|87.44|2.42|49.86|
> |GyroAtt-SPSD|AIM|6.78|123.44|2.37|65.30|
> |GyroAtt-SPSD|LEM|6.54|116.49|2.34|64.23|
> |GyroAtt-SPSD|LCM|5.71|103.57|2.16|58.50|

---

> ### Author Response · Authors · 2024-11-24
> **Common Response (3/3)**
>
> **CQ#4. More focused experimental comparisons with MAtt and GDLNet.** $\textcolor{orange}{Q2Fj (R3)}$, $\textcolor{purple}{nEzp (R4)}$
>
> Beyond computational complexity and runtime analysis, we have conducted a detailed comparison of the proposed GyroAtt with MAtt [b] and GDLNet [c] across the following  two aspects:
> - **Ablation on Attention Blocks**: To evaluate the significance of our Gyro Attention block, we replace it with the designed attention blocks in MAtt and GDLNet. As shown in Tab.D, this replacement leads to a decrease in the classification ability of GyroAtt-SPD and GyroAtt-Gr on the BNCI2014001 and BNCI2015001 datasets.
> - **Ablation on GyroAtt Components**: We further evaluate the contributions of Gyro Homomorphism and nonlinear activation function by replacing them with equivalent layers from MAtt and GDLNet, including BiMap, FrMap, and ReEig. As shown in Tab. E and F, the aforementioned replacements result in the accuracy degradation of GyroAtt-SPD and GyroAtt-SPSD on the two used datasets.
>
> Table D: Accuracy (\%) comparison of GyroAtt under different attention blocks, where Ab-MAtt and Ab-GDLNet denote the attention blocks in MAtt and GDLNet, respectively.
> |Methods|BNCI2014001 Inter-session|BNCI2014001 Inter-subject|BNCI2015001 Inter-session|BNCI2015001 Inter-subject|
> |-|-|-|-|-|
> |GyroAtt-SPD (Ab-MAtt)|72.4±8.5|51.9±14.7|84.6±7.8|76.5±10.1|
> |GyroAtt-SPD |**75.4±7.4**|**53.1±14.1**|**86.2±4.5**|**77.9±13.0**|
> |GyroAtt-Gr (Ab-GDLNet)|69.2±9.7| 50.3±5.1|83.9±11.2| 73.5±13.7|
> |GyroAtt-Gr|**72.5±7.3**|**52.1±14.2**|**85.0 ±7.7**|**75.3±13.7**|
>
> Table E: Accuracy (\%) comparison of GyroAtt-SPD under different layer combinations.
> |Transformation|Activation|BNCI2014001 Inter-session|BNCI2014001 Inter-subject|BNCI2015001 Inter-session|BNCI2015001 Inter-subject|
> |-|-|-|-|-|-|
> |BiMap|Power|74.0±6.5|52.3±15.0|85.2±7.2|77.2±13.2|
> |Homomorphisms|ReEig|75.1±6.3|52.6±14.2|85.9±5.3|76.4±12.8|
> |BiMap|ReEig|73.6±6.8|52.2±15.2|85.4±7.8|76.8±13.0|
> |Homomorphisms|Power|**75.4±7.4**|**53.1±14.1**|**86.2±4.5**|**77.9±13.0**|
>
> Table F: Accuracy (\%) comparison of GyroAtt-SPSD under different layer combinations.
> |Transformation|Activation|BNCI2014001 Inter-session|BNCI2014001 Inter-subject|BNCI2015001 Inter-session|BNCI2015001 Inter-subject|
> |-|-|-|-|-|-|
> |FrMap| Power|68.9±6.9|51.2±12.9|82.3±6.2|65.8±13.1|
> |Homomorphisms|ReEig|72.3±6.9| 49.6±13.3|84.9±6.2|74.1±12.3|
> |FrMap|ReEig|68.8±7.2|50.7±13.8|81.6±6.1|72.9±13.3|
> |Homomorphisms|Power|**72.9±6.2**|**52.4±15.6**|**85.3±5.3**|**76.0±14.1**|
>
>
> We explain the merits of GyroAtt from the following two perspectives:
> - **Respecting Gyro Algebraic Structure via Gyro Homomorphisms**: The transformation layers designed in GyroAtt leverage gyro homomorphisms, which inherently preserve the gyro algebraic structure, including gyro addition $\oplus$ and gyro scalar multiplication $\otimes$. These transformations, rooted in Riemannian operators, ensure that GyroAtt inherently respects the Riemannian geometry of the manifold. In contrast, the transformation layers in MAtt and GDLNet, such as BiMap and FrMap, only partially adhere to Riemannian geometry. While these layers enforce manifold constraints, they lack isometric properties, potentially distorting the intrinsic geometric structure. For example, the distance $d(S_1, S_2)$ between two SPD matrices $S_1$ and $S_2$ may not be preserved after transformation, i.e., $d(\mathrm{BiMap}(S_1), \mathrm{BiMap}(S_2)) \neq d(S_1, S_2)$. This limitation undermines their ability to fully respect and retain the original geometric properties of the data, which can negatively impact performance.
> - **Nonlinear Activation through Matrix Power $\sigma(\cdot)$**: Existing SPD networks often employ the ReEig function for nonlinear activation, focusing primarily on maintaining numerical stability and ensuring SPD-ness. However, ReEig operates purely as a numerical adjustment, lacking any intrinsic connection to the underlying Riemannian geometry. In contrast, GyroAtt utilizes a matrix power-based nonlinear activation function that directly manipulates the Riemannian metric while preserving the SPD structure. This approach not only aligns with the manifold's latent geometry but also offers greater flexibility. As illustrated in [d] (Fig. 1), the matrix power operation enables transitions between Riemannian metrics. For instance, as the parameter $\theta$ approaches zero, the AIM gradually transitions into the LEM, showcasing the adaptability of our method.
>
> **References**
>
> > [a] Attention is All You Need. In NeurIPS 2017.
> >
> > [b] MAtt: A Manifold Attention Network for EEG Decoding. In NeurIPS 2022.
> >
> > [c] A Grassmannian Manifold Self-Attention Network for Signal Classification. In IJCAI 2024.
> >
> > [d] RMLR: Extending multinomial logistic regression into general geometries. In NeurIPS 2024.

---

### Meta-Review · Area_Chair_fSxe · 2024-12-17

**Metareview:**

The current submission extends attention mechanisms to gyrovector spaces instantiated by three matrix manifolds and tested against EEG classification tasks. The generalization is meaningful as gyrovector space has recently been imported into deep learning as a powerful tool. The reviewers agreed that the direction the authors have taken is promising, acknowledged the empirical improvement, and noted that the readability has been improved during the rebuttal phase.

The reviewers still have concerns regarding the presentation and the soundness of the contribution, based on which the paper is not suitable for publication in its current form and would benefit from further iterations.

The early section could be better motivated by explaining what aspects of gyrovector spaces lead to their possibility of addressing the limitations of existing attention mechanisms. As a technique grounded in deep theories, its advantages should not be based solely on the empirical improvement but rather on why these improvements are expected. The authors are encouraged to position the work among the existing work of gyrovector spaces and their applications in deep learning, and use the space to highlight the main contribution in sections 4 and 5. The proposed attention framework relies on both the gyro-structure for performing feature transformations and the Riemannian structure for computing the geodesic distances, and how they interplay and contribute to the experimental success requires further empirical study and discussions.

**Additional Comments On Reviewer Discussion:**

All reviewers participated either in the rebuttal or in the final discussion.

Two reviewers zEZt and nEzP are weighted down due to low confidence and relatively short reviews, and that they did not comment during the reviewers-AC discussion.

One of the the reviewers bQUv raised multiple clarity issues, which is addressed by the authors in the rebuttal. However the reviewer is still concerned on the lack of motivation, and confirmed their rating as "not enough for acceptance".

Upon reading the paper, I reconfirmed the weakness and other issues mentioned in the above meta-review.

The other reviewer Q2Fj, which gives the highest rating 8, agrees on the weaknesses raised during the discussion.

---

### Decision · Program_Chairs · 2025-01-22

Reject